

# Sensitivity of ocean biogeochemistry to the iron supply from the Antarctic ice sheet explored with a biogeochemical model

Renaud Person[1], Olivier Aumont[1], Gurvan Madec[1], Martin Vancoppenolle[1], Laurent Bopp[2], and Nacho Merino[3]

[1]Laboratoire d'Océanographie et du Climat: Expérimentations et Approches Numériques (LOCEAN), IPSL, Sorbonne Université, Paris, 75005, France
[2]Laboratoire de Météorologie Dynamique (LMD), IPSL, École Normale Supérieure – PSL Research University, CNRS, Sorbonne Université, Ecole Polytechnique, Paris, 75005, France
[3]Université Grenoble Alpes, Institut des Geosciences de l'Environnement (IGE), CNRS, IRD, Grenoble, 38000, France

**Correspondence:** Renaud Person (renaud.person@locean-ipsl.upmc.fr)

**Abstract.** Iron (Fe) delivery by the Antarctic Ice Sheet (AIS) through ice shelf and iceberg melting enhances primary productivity in the largely iron-limited Southern Ocean (SO). To explore this fertilization capacity, we implemented a simple representation of the AIS iron source in the global ocean biogeochemical model NEMO-PISCES. We evaluated the response of Fe, surface chlorophyll, primary production and carbon export to the magnitude and hypothesized vertical distributions of the AIS Fe fluxes. Surface Fe and chlorophyll concentrations are increased up to 25 % and 12 %, respectively, over the whole SO. The AIS Fe delivery is found to have a relatively modest impact on SO primary production and C export which are increased by 0.063 ± 0.036 PgC yr$^{-1}$ and 0.028 ± 0.016 PgC yr$^{-1}$, respectively. However, in highly fertilized areas, primary production and C export can be increased by up to 30 % and 42 %, respectively. Icebergs are predicted to have a much larger impact on Fe, surface chlorophyll and primary productivity than ice shelves in the SO. The response of surface Fe and chlorophyll is maximum in the Atlantic sector, northeast of the tip of the Antarctic Peninsula, and along the East Antarctic coast. The iceberg Fe delivery below the mixed layer may, depending on its assumed vertical distribution, fuel a non-negligible subsurface reservoir of Fe. The Fe supply is effective all year round and seasonal variations in iceberg melting have regional impacts which are almost negligible for annual-mean primary productivity and C export at the scale of the SO.

## 1   Introduction

Iron (Fe) is a vital micronutrient for phytoplankton photosynthesis and marine life. While being the fourth most abundant element in the continental crust (Wedepohl, 1995), Fe is present at extremely low concentrations in most of the oceans. In the SO, the largest High Nutrient Low Chlorophyll (HNLC) region, this trace metal exerts with light a strong limitation on primary productivity (Boyd et al., 2007; Coale et al., 2004; Martin et al., 1990; Smetacek, 2001). Iron supply therefore modulates the



intensity of the biological carbon pump in the SO (Blain et al., 2007; Bowie et al., 2001; Boyd et al., 2007) and possibly plays a key role on glacial-interglacial carbon-cycle regulation of climate (Martin, 1990).

Several sources contribute to the Fe pool in the SO: atmospheric dust deposition (Raiswell et al., 2016; Tagliabue et al., 2009), sediment resuspension and dissolution (Borrione et al., 2014; de Jong et al., 2013; Dulaiova et al., 2009; Tagliabue et al.,
2009), hydrothermal activity (Tagliabue et al., 2010), icebergs calving and melting (Duprat et al., 2016; Lin et al., 2011; Smith et al., 2007), ice shelves (Gerringa et al., 2012; Herraiz-Borreguero et al., 2016) Herraiz-Borreguero et al. (2016), and sea ice (Lancelot et al., 2009; Lannuzel et al., 2007, 2010, 2016). Modeling studies have highlighted the different levels of significance of these Fe sources to sustain primary productivity in the SO (Borrione et al., 2014; Death et al., 2014; Lancelot et al., 2009; Laufkötter et al., 2018; Tagliabue et al., 2009, 2014a; Wadley et al., 2014; Wang et al., 2014). Nevertheless, large uncertainties
remain in their fertilization capacity due to an important lack of data, hampering their integration in biogeochemical and climate models (Tagliabue et al., 2016).

Among the Fe sources in the SO, icebergs and ice shelves have been largely overlooked in ocean biogeochemical models. For instance, none of the models participating to the FeMIP exercise includes these glacial iron sources (Tagliabue et al., 2016) while observations estimate the mean flux of potentially bioavailable Fe from SO icebergs to span 1 to 3 orders of magnitude
higher than from dust deposition (Raiswell et al., 2016; Shaw et al., 2011) ranging from 3.2 to 25 Gmoles yr$^{-1}$ for icebergs and from 0.0 to 0.02 Gmoles yr$^{-1}$ for atmospheric dust (Raiswell et al., 2016). The few modeling studies scaled the contribution of the AIS iron source in the same order of magnitude as atmospheric dust (Death et al., 2014; Lancelot et al., 2009) or one order of magnitude higher (Laufkötter et al., 2018; Wadley et al., 2014) but with a larger uncertainty in the biological response to its fertilization effect. Thus, the iceberg Fe source is estimated to increase the SO primary production by 6 % to 10 % in Wadley
et al. (2014) while Death et al. (2014) evaluated the iceberg and subglacial contribution up to 40 %. Recently, Laufkötter et al. (2018) estimated, in a preindustrial context, the AIS iron source to sustain 30 % of the marine particle export production in the SO consequently reducing by 30 % the carbon outgassing in this region.

Icebergs and ice shelves contain higher Fe concentrations than seawater (de Baar et al., 1995; Herraiz-Borreguero et al., 2016; Lin et al., 2011; Shaw et al., 2011), mainly as lithogenic material from glacial sediments (Hopwood et al., 2017; Raiswell
et al., 2006; Shaw et al., 2011). The melting of icebergs and ice shelves releases Fe to seawater through finely ground rocks as particulate, dissolved, and potentially dissolvable forms (Hawkings et al., 2014; Herraiz-Borreguero et al., 2016; Hodson et al., 2017; Raiswell et al., 2008, 2016), fueling surface waters and the water column (De Jong et al., 2015; Lin et al., 2011). Fe in glacial sediments was long considered to be unavailable to phytoplankton. Raiswell et al. (2006) showed that glacial sedimentary Fe contains nanoparticulate Fe of which a small fraction can be biogeochemically reactive and potentially
bioavailable to phytoplankton. The iron fertilization capacity of icebergs has been evidenced from *in situ* observations (Lin et al., 2011; Smith et al., 2007) and hotspots of primary productivity have been observed by satellites in the wake of drifting icebergs (Duprat et al., 2016; Schwarz and Schodlok, 2009). In coastal regions, the under ice shelf delivery of bioavailable Fe can also be significant to sustain primary productivity as estimated in the Amundsen Sea (Gerringa et al., 2012) and in the Prydz Bay (Herraiz-Borreguero et al., 2016). However, the mean supply of the bioavailable Fe fraction from icebergs and
ice shelves is difficult to quantify because of the heterogeneous nature of the Fe distribution in these sources (Hopwood et al.,



2017; Raiswell et al., 2016). Until recent years, very few data were available. Estimates of iceberg Fe fluxes were based on only 6 samples (Raiswell et al., 2008) and, to our knowledge, no representative data are available for ice shelves. New observations increased the set of iceberg data to about 50 glacial samples (Raiswell et al., 2016), offering the opportunity to better constrain the Fe supply from the AIS to seawater and its effect on primary productivity in biogeochemical models.

Quantifying the contribution of the AIS to the Fe pool in the SO is of great interest for marine biogeochemistry as this source may be influenced by global warming. Indeed, the SO is a large sink of anthropogenic carbon (Khatiwala et al., 2013; Sabine et al., 2004) whose physical environment is evaluated to be severely affected by climate change (Rintoul et al., 2018). The AIS has already lost a significant amount of its mass over the period 1992-2017. The total loss of ice mass is of 2,720 ± 1,390 billion tons, and is particularly strong in West Antarctica and in the Antarctic Peninsula region, where annual melting rates

have increased by factors of 3 and 5, respectively (The IMBIE team, 2018). In a business as usual scenario, the glacial coverage in Antarctica is estimated to be massively altered with a possible 23 % reduction of the ice shelf volume by 2070 (DeConto and Pollard, 2016; Rintoul et al., 2018). The projected decline of the AIS will increase the release of Fe from icebergs and ice shelves in the SO with possible significant impacts on marine productivity and biogeochemical cycles.

In this study, we evaluate the AIS impacts on Fe concentrations and marine primary productivity in the SO and investigate

their sensitivity to the main characteristics of the iron delivery from icebergs and ice shelves. Firstly, we focus on the magnitude of the AIS Fe supply. For this purpose, different soluble fractions of sedimentary Fe were assumed in the ocean biogeochemical model NEMO-PISCES, associated with recent iceberg and ice shelf freshwater flux estimates. Secondly, because the distribution of released Fe from icebergs along the water column is largely undocumented, we investigated several possible vertical distributions of iceberg Fe delivery to seawater to encompass this large uncertainty. The effects of the seasonal variations in

the iceberg Fe supply are evaluated against an annual mean climatology of the iceberg Fe fluxes. We also evaluated the relative contributions of ice shelves and icebergs to the SO Fe pool.

## 2   Method

### 2.1   NEMO-PISCES model description

We used the hydrodynamical and biogeochemical model NEMO-PISCES version 3.6 (Madec, 2008). This modeling platform

is based on the ocean dynamical core OPA (Madec, 2008), the marine biogeochemistry model PISCES-v2 (Aumont et al., 2015), and the Louvain-La-Neuve sea ice model LIM3 version 3.6 (Rousset et al., 2015). We used a global configuration of NEMO-PISCES at 1° horizontal resolution of an isotropic mercator grid with a local meridional refinement up to 1/3° at the equator. The vertical grid follows a partial step z-coordinate scheme and has 75 levels with 25 levels in the upper 100 m. Lateral mixing is computed along isoneutral surfaces (Madec, 2008). Mesoscale eddy-induced turbulence follows the

Gent and Mc Williams (1990) parametrization and vertical mixing is parameterized using the turbulent kinetic energy scheme (Blanke and Delecluse, 1993) as modified by Madec (2008). The biogeochemical model PISCES simulates two phytoplankton functional types (diatoms and nanophytoplankton), two zooplankton size classes (microzooplankton and mesozooplankton), the biogeochemical cycles of five limiting nutrients ($NO_3$, $PO_4$, $NH_4$, $Si(OH)_4$, and Fe), dissolved oxygen, dissolved inorganic





carbon, total alkalinity, dissolved organic matter, small and large organic particles. Different external sources of Fe are included: atmospheric dust deposition, sediment mobilization, rivers, and sea ice. The implementation of these iron sources in NEMO-PISCES is fully described in Aumont et al. (2015).

## 2.2 Modeling the Antarctic ice sheet Fe supply

To represent the AIS iron supply to seawater in our model, we used recent freshwater flux climatologies of icebergs and ice shelves based on Depoorter et al. (2013). The modeled annual mean freshwater flux from the AIS is estimated to ~2790 Gt yr⁻¹ partitioned into a liquid and a solid phase of about the same magnitude with an annual release of ~1439 Gt yr⁻¹ from ice shelves and of ~1351 Gt yr⁻¹ from icebergs. The climatology of the coastal runoff estimate of Antarctic ice shelves is assumed to be a steady freshwater flux throughout the year. The ice shelf freshwater flux is homogeneously distributed along the water

column between the base and the grounding line of ice shelves using the prescribed meltwater flux parameterization of Mathiot et al. (2017). This parameterization has been developed to represent the unresolved overturning circulation in cavities beneath ice shelves where melting mainly occurs (Depoorter et al., 2013; Herraiz-Borreguero et al., 2016; Mathiot et al., 2017). For icebergs, we used a model-based seasonal climatology of iceberg melting over the SO (Fig. 1) from Merino et al. (2016). The monthly climatology distribution of freshwater flux from icebergs has been estimated using an improved version of the

Lagrangian iceberg model NEMO-ICB (Marsh et al., 2015) coupled to a 1/4° global configuration of NEMO (Merino et al., 2016). The ocean model was forced by a climatological repeated-year atmospheric forcing based on ERA-interim and by recent estimates of Antarctic freshwater (Depoorter et al., 2013).

    The iceberg-hosted sediment content is poorly constrained by observations and is estimated to range from 0.4 to 1.2 g L⁻¹ (Anderson et al., 1980; Shaw et al., 2011). To simulate the Fe fluxes delivered by melting icebergs and ice shelves in the SO,

we associated to the freshwater flux climatologies a sediment content of 0.5 g L⁻¹ as used in Raiswell et al. (2006) and Death et al. (2014) assuming, as a crude assumption, that sediment content in icebergs and ice shelves are roughly equivalent. The mean content of labile Fe in iceberg-hosted sediments, mainly in the form of ferrihydrite, has been recently estimated to range from 0.03 % to 0.194 % with a mean value of 0.076 % (Raiswell et al., 2016). Shaw et al. (2011) estimated a range of labile Fe of 0.04 % to 0.4 % for free-drifting icebergs in the Weddell Sea. In our study, we set the mean sediment content in ferrihydrite

to be 0.1 %. The fraction of ferrihydrite that can be biologically available as Fe nanoparticles (i.e. the soluble fraction of ferrihydrite) is assumed to be of 10 % (Death et al., 2014; Raiswell et al., 2008). In order to account for the uncertainty of the bioavailable fraction of glacial Fe (Boyd et al., 2012), we used a solubility within a range of 1 % to 10 % which corresponds to a total annual Fe flux of 0.25 to 2.5 Gmoles yr⁻¹ (Table 1). This range is relatively similar to other modeling studies (Death et al., 2014; Laufkötter et al., 2018). The modeled iceberg Fe fluxes are in the lower range of previously published estimates

based on observations (Raiswell et al., 2008, 2016; Shaw et al., 2011). To our knowledge, no data allow to constrain the ice shelves Fe fluxes as the Antarctic estimates from Hawkings et al. (2014) are extrapolated from Greenland ice sheet data.





## 2.3 Experimental design

We designed 9 model experiments with different Fe solubilities for both ice shelves and icebergs, and different vertical distributions of delivered Fe from icebergs (Table 2). For consistency with the climatological forcing of the Antarctic freshwater release, all these experiments are run in a climatological setup using the CORE-I normal year atmospheric forcing (Griffies

et al., 2009) and are initialized from a 120 years long spin up simulation. They all include external sources of Fe from dust, sediments, sea ice, and rivers even though the latter does not contribute to the Fe pool in the SO. Each experiment is run for 20 years to achieve a sufficient equilibrium state for the Fe cycle in the framework of our sensitivity study.

The control experiment (CTL) is used as a reference run in the rest of the study and does not take into account any iron source from the AIS. Figure 2 shows the annual mean distribution of surface Fe concentrations over the SO simulated by

the CTL experiment. This distribution is contrasted with regions showing high surface Fe concentrations in coastal regions around the Antarctic continent and in the surrounding waters of SO islands such as South Georgia, the Crozet archipelago, the Kerguelen Plateau, and with large areas in the open ocean displaying very low values. The modeled surface distribution of Fe concentrations reflects the main contribution of sediments among the different external iron sources actually implemented in the standard version of the PISCES model (Aumont et al., 2015). The Fe distribution of the NEMO-PISCES model has been

validated at the global scale in Tagliabue et al. (2016) and over the SO in Person et al. (2018) showing reasonable performances compared to available data (Tagliabue et al., 2012).

Three different solubilities of Fe from icebergs and ice shelves are tested in the SOLUB1, SOLUB5, and SOLUB10 experiments, imposing 1 %, 5 %, and 10 %, respectively. The corresponding annual Fe fluxes amount to 0.25, 1.25, and 2.5 Gmoles yr$^{-1}$, respectively, with similar contributions from both glacial sources (Table 1). The ISF, ICB-SURF, ICB-ML, ICB-KEEL,

and ICB-ANNUAL experiments have an iceberg and ice shelf Fe solubility of 5 % as in the SOLUB5 experiment. The ISF experiment only includes the Fe source from ice shelves in order to assess its contribution against icebergs.

Different vertical distributions of the iceberg Fe fluxes have been tested. In the SOLUB1, SOLUB5, SOLUB10 and ICB-ANNUAL experiments, Fe is homogeneously released from icebergs over the top 120 m of the water column. This value corresponds to the average depth of the submerged part of the five class sizes of icebergs modeled by the NEMO-ICB model

(Marsh et al., 2015) and computed by applying the formulation of Rackow et al. (2017) to the average thickness of the modeled icebergs. In the ICB-SURF experiment, the whole iceberg Fe supply is released at the surface, i.e. in the first vertical level of our model which is 1 m thick. In the ICB-KEEL experiment, this flux is released at ~120 m depth, i.e. at the mean depth of the keel of modeled icebergs. The ICB-KEEL experiment is set up to evaluate the contribution of a theoretical distribution of iceberg Fe fluxes delivered only at the base of icebergs. The ICB-KEEL experiment can be seen as the mirror experiment of

the ICB-SURF experiment keeping in mind that this distribution is most probably unrealistic, the buoyancy effect tending to upwell the iceberg meltwater to the surface (Smith et al., 2007). In order to evaluate the role played by the iceberg Fe fluxes distributed below the mixed layer (ML), that is, the fraction not directly available for surface primary productivity, we designed the ICB-ML experiment where this fraction is removed. Thus, the iceberg Fe fluxes in the ICB-ML experiment are distributed along the water column, i.e. until 120 m depth, as in the SOLUB5 experiment, but the iceberg Fe flux values below the MLD





are set to zero unlike in the SOLUB5 experiment. Finally, in the ICB-ANNUAL experiment, an annual mean climatology of the iceberg Fe fluxes is used instead of the monthly climatology to evaluate the impact of the seasonal variability in the supply of Fe from icebergs in the SO.

## 3 Results

### 3.1 Contribution of the Antarctic ice sheet to the spatial distribution of Fe

#### 3.1.1 Sensitivity to the magnitude of the Antarctic ice sheet Fe supply

The uncertainty in the magnitude of the AIS Fe source is estimated to span, at least, 1 order of magnitude (Table 1). We evaluated the impact of this range on the spatial distribution of Fe in the SO by imposing three different soluble fractions of Fe: 1 %, 5 %, and 10 % (Table 2). The Fe supply from the AIS increases the Fe concentrations in the first 120 m in the SOLUB1, SOLUB5, and SOLUB10 experiments compared to the CTL experiment, respectively, the surface anomaly increasing with the Fe solubility (Fig. 3).

Globally, higher surface Fe concentrations are simulated in coastal regions all around the Antarctic continent. The most noticeable Fe anomaly is a marked plume northeast of the Antarctic Peninsula that expands until 50° S in the Atlantic sector and reaches the western sector of the Indian Ocean (Fig. 3c-3f). The spatial extent of the Fe anomalies becomes larger as the Fe solubility increases, particularly in the Atlantic sector and in the Ross Sea which appear to be the offshore areas that are the most greatly influenced by the AIS Fe source. The SOLUB1 experiment simulates a moderate impact with an annual mean surface Fe concentrations higher by ∼0.015 nmol L$^{-1}$ over the SO, south of 50° S, relative to the CTL experiment, i. e. 3 % more. The supply in the Atlantic plume increases the surface Fe concentrations by up to 0.16 nmol L$^{-1}$ in summer (Fig. 3a). The highest Fe values are found in winter along the coasts of the Ross Sea and of the Amundsen Sea with surface Fe anomalies that reach 1 nmol L$^{-1}$ (Fig. 3a and 3b). In the SOLUB5 experiment, the contribution of the AIS Fe source is more significant with a mean surface Fe concentration that is ∼0.07 nmol L$^{-1}$ higher, i.e. 13 % more than in the CTL experiment (Fig. 3c and 3d). The Atlantic plume is clearly marked and extends further eastward until 10° E with surface Fe concentrations in summer up to ∼0.8 nmol L$^{-1}$ higher (Fig. 3c). Along the Antarctic coasts, the AIS supply in winter increases the surface Fe anomalies by up to 3.8 nmol L$^{-1}$ particularly in the Indian and Pacific sectors as well as in the Ross Sea until the Amundsen Sea. Two additional plumes emerge: a large one north of the Ross Sea and a smaller one in the vicinity of South Georgia (Fig. 3c and 3d). The SOLUB10 experiment strengthens the seasonal and spatial patterns of the surface Fe anomalies simulated in SOLUB5 with extensive Fe anomalies in the Atlantic plume, in the Ross Sea, in the Weddell Sea, and all along the Antarctic coasts (Fig. 3e and 3f). Over the SO, south of 50° S, the annual mean Fe concentrations are ∼0.13 nmol L$^{-1}$ higher than in the CTL experiment, an increase of 24 %. The surface Fe concentrations in the Atlantic plume and along the Antarctic coasts are up to ∼1.5 nmol L$^{-1}$ and ∼6.3 nmol L$^{-1}$ higher, respectively. With a Fe solubility of 10 %, the SOLUB10 experiment predicts an important contribution of the AIS source to the SO Fe pool (Fig. 3e), even larger near the coasts in winter (Fig. 3f).





The AIS significantly alters the surface Fe concentrations both in summer and winter (Fig. 3). The spatial patterns between these two seasons exhibit noticeable differences. In summer, surface Fe anomalies are marked and intense whereas, in winter, they extend over larger areas and are more diffuse (Fig. 3c-3f) showing lower maximum values but having higher mean levels. These seasonal differences reflect two different dynamics in the supply of Fe from the AIS and its subsequent loss from the

surface. In summer, the release of Fe associated to more intense iceberg freshwater fluxes drives surface Fe concentrations to high values. Environmental conditions are favorable for phytoplankton growth and the intense biological activity efficiently consumes the supplied Fe preventing it to be transported over large distances, especially in iron limited areas. In winter, biological activity is much weaker due to the strong light limitation and the delivered Fe from the AIS can be advected further away. Furthermore, in winter, deep mixed layer entrained to the surface Fe released below the euphotic zone that thus escapes

summer consumption by phytoplankton. This unconsumed fraction is also advected over significant distances by the intense ocean circulation in the SO. This explains the much sharper gradients simulated in summer, particularly noticeable in the SOLUB5 and SOLUB10 experiments.

### 3.1.2   Sensitivity to vertical distributions of the iceberg Fe supply

It is well established that ice shelf meltwater is injected at depth into the ocean (Depoorter et al., 2013; Mathiot et al., 2017),

the basal melting being driven by the properties of water masses that enter the ocean cavities underneath ice shelves (Jacobs et al., 1992). While the iceberg Fe supply has been evidenced by *in situ* observations (De Jong et al., 2015; Lin et al., 2011; Shaw et al., 2011), almost nothing is known to our knowledge of where the Fe delivery mostly occurs along the immersed part of icebergs and where this input is mainly available to phytoplankton. Nonetheless, FitzMaurice et al. (2017) recently pointed out that the nonlinear response of iceberg melting leads to meltwater injected near the surface or mixed at depth depending

on whether the flow velocity is weak or strong, respectively. Here we evaluate the impacts of four different theoretical vertical distributions of iceberg Fe fluxes on surface Fe concentrations over the SO as well as on vertical profiles of Fe in the upper 300 m of a large area highly fertilized by the AIS northeast of the Antarctic Peninsula (36° W-56° W, 58° S-63° S). For this purpose, we compare the ICB-SURF, ICB-ML, and ICB-KEEL experiments against the SOLUB5 experiment (Fig. 4).

The surface distribution of the iceberg Fe fluxes in the ICB-SURF experiment results in a large excess of surface Fe concen-

trations in summer, all over the SO, compared to the volume distribution applied in the SOLUB5 experiment (Fig. 4a). This excess is important with surface Fe concentrations reaching values higher by up to 27 nmol $L^{-1}$. In the ICB-SURF experiment, the iceberg Fe supply in the mixed layer is maximum and is not sensitive to the depth of the mixed layer. By contrast, when the Fe flux is distributed over the upper 120 m (SOLUB5), the shallow pycnocline in summer severely limits the iceberg Fe supply in the mixed layer, most of this supply being injected below the MLD. The differences between both experiments in winter are

significantly less marked with large patterns of positive and negative differences in surface Fe concentrations highlighting the role played by the interactions between the seasonal variations of the MLD and the injection of Fe at depth (Fig. 4b). When the MLD is deeper than 120 m, the ICB-SURF experiment simulates slightly lower Fe concentrations, up to ∼0.04 nmol $L^{-1}$ lower than in SOLUB5 in the Atlantic sector south of 60° S and, globally, all around the Antarctic coasts. The boundary zone between positive and negative values in the Atlantic sector is driven by the interplay between MLD shallower than 120 m (Fig. 5) and



the oceanic circulation resulting in Fe concentrations up to 1.5 nmol L$^{-1}$ higher in the ICB-SURF experiment than in SOLUB5. Finally, higher Fe concentrations are simulated in ICB-SURF in localized areas in the Ross, Amundsen, and Bellingshausen Seas, and near the coasts of the Indian sector between 70° E and 85° E but without clear correlation with the MLD.

The vertical profiles of Fe in the highly fertilized area of the Atlantic plume northeast of the Antarctic Peninsula (36° W-56° W, 58° S-63° S) illustrate the different dynamic in the seasonal supply of Fe to the upper ocean in both experiments (Fig. 6). In summer, these vertical profiles are very different (Fig. 6a). In the mixed layer, Fe concentrations are higher in ICB-SURF than in SOLUB5 by a factor of 2.2. Below the MLD, the ICB-SURF experiment simulates Fe concentrations that decrease strongly until 70 m. In the SOLUB5 experiment, Fe concentrations increase significantly from below 30 m until 120 m and from there, decrease until 150 m depth. Below 150 m depth, both experiments converge to the same vertical profile of Fe.

In winter, the vertical profiles are qualitatively similar in the upper 300 m. Yet, the ICB-SURF experiment displays a smaller vertical gradient in the upper 150 m than in SOLUB5 (Fig. 6b). The scarcity of the data makes it challenging to discriminate whether the ICB-SURF experiment or the SOLUB5 experiment simulates a realistic vertical distribution of Fe.

    The ICB-ML experiment allows to quantify the influence of the iceberg Fe supplied below the MLD, i.e. the importance of the non-directly available fraction of the iceberg Fe source, on the spatial distribution of Fe over the SO. Surface Fe concen-

trations in the ICB-ML experiment are lower than in the SOLUB5 experiment in both seasons and over the whole SO (Fig. 4c and 4d). Surface Fe values in summer and winter are up to ∼0.55 nmol L$^{-1}$ and ∼0.4 nmol L$^{-1}$ lower, respectively, than in the SOLUB5 experiment. This comparison suggests that the Fe fraction delivered by icebergs below the MLD is not completely scavenged and constitutes a Fe pool that can supply surface waters in Fe as soon as the mixed layer deepens.

    The seasonal evolution of the vertical Fe profiles supports the important role of the subsurface additional pool of Fe due to

iceberg melting (Fig. 6). In summer, the SOLUB5 experiment has a Fe concentration in the mixed layer ∼0.1 nmol L$^{-1}$ higher than in the ICB-ML experiment (Fig. 6a). Below the MLD, Fe concentrations in SOLUB5 display a local maximum between 30 and 150 m. In contrast, the ICB-ML experiment does not simulate this local maximum in summer. In winter, Fe profiles in SOLUB5 and ICB-ML are qualitatively almost similar except that iron levels in ICB-ML are about 0.06 nmol L$^{-1}$ higher (Fig. 6b). This comparison illustrates that the Fe released by icebergs in summer below the MLD may represent a significant

subsurface reservoir that can feed in Fe the surface layer by intraseasonal events such as storms (Nicholson et al., 2016; Swart et al., 2015), by strong meso- and sub-mesoscale activities (Rosso et al., 2016; Swart et al., 2015) as well as by deep mixing in winter (Tagliabue et al., 2014b).

    The iceberg Fe supply at depth in the ICB-KEEL experiment shows a significant decrease in surface Fe concentrations compared to the SOLUB5 experiment in both seasons (Fig. 4e and 4f). In summer, surface Fe concentrations are up to ∼2.8

nmol L$^{-1}$ lower than in the SOLUB5 experiment in the Atlantic plume and all around the Antarctic coasts (Fig. 4e). In winter, the difference is weaker than in summer with surface Fe concentrations up to ∼0.8 nmol L$^{-1}$ lower than in the SOLUB5 experiment (Fig. 4b). Moreover, the spatial differences between both experiments in winter in the open ocean are less widespread than in the ICB-ML experiment (Fig. 4d) where the iceberg fertilization effect is less effective, south of the Atlantic plume and, more generally, south of 60° S offshore of the Antarctic coasts. While low, the supply of Fe from icebergs at depth can have a large

area of influence on surface Fe concentrations in winter.





The vertical profile of Fe in the Atlantic plume presents a marked peak at 120 m depth, which corresponds to the depth at which Fe is being released from icebergs melting. At that depth, iron concentrations reach 1.2 nmol L$^{-1}$ in summer, which is 0.5 nmol L$^{-1}$ higher than in the SOLUB5 experiment (Fig. 6a). In the upper layer, Fe concentrations are lower by ∼0.2 nmol L$^{-1}$ than in the the SOLUB5 experiment and almost equal to the CTL experiment. The vertical gradient is the strongest of all the

experiments. In winter, surface Fe concentrations in the mixed layer are ∼0.09 nmol L$^{-1}$ lower than in the SOLUB5 experiment and slightly higher than in the CTL experiment (Fig.65b). The vertical gradient between the surface and 120 m depth remains stronger than in any other experiments but the difference is weaker. Below 120 m and down to about 200 m, differences with the other experiments are significantly smaller than in summer. These results show that a predominant supply of Fe at the base of icebergs will generate an important subsurface reservoir of iron that can be entrained to the surface by the deepening of

the MLD. The role of the subsurface reservoir of Fe is pointed out to be critical to sustain the iron supply to surface waters (Tagliabue et al., 2014b).

### 3.1.3  Sensitivity to the seasonal variations of the iceberg Fe supply

The variations of the AIS Fe fluxes due to the seasonal variability of calving and melting of icebergs (Fig. 1a) impact the seasonal cycle of Fe over the SO. To assess to what extent these variations are significant for the Fe pool in the SO, we compare

the ICB-ANNUAL experiment to the SOLUB5 experiment. Globally, over the whole SO, the surface Fe concentrations in the ANNUAL experiment and the SOLUB5 experiment are increased by 9 % and 13 % in summer and by 15 % and 13 % in winter, respectively, relative to the CTL experiment. Imposing an annual mean iceberg supply of Fe also leads to differences in the spatial distribution of Fe. In ICB-ANNUAL, surface Fe concentrations in summer are lower in the Atlantic sector and all around the Antarctic coasts than in the SOLUB5 experiment with values up to ∼1.9 nmol L$^{-1}$ lower (Fig. 4g). On the

other hand, some other areas such as downstream of South Georgia, in the Weddell Sea, and in the Ross Sea are predicted to have higher Fe concentrations. In the Weddell Sea, along the east coasts of the Antarctic Peninsula, the Fe values in the ICB-ANNUAL experiment are up to 0.2 nmol L$^{-1}$ higher than in the SOLUB5 experiment. In winter, an opposite spatial pattern is simulated. Surface Fe concentrations in the Atlantic plume and along the east coasts are up to ∼0.75 nmol L$^{-1}$ higher in the ICB-ANNUAL experiment whereas offshore of 80° E, downstream of South Georgia, in the Weddell Sea, and in the

Ross Sea, these concentrations are up to ∼0.45 nmol L$^{-1}$ lower (Fig. 4h). When looking at the vertical Fe distribution in the Atlantic plume, vertical profiles in summer have almost the same shape in both experiments. However, Fe concentrations in ICB-ANNUAL are lower by ∼0.1 nmol L$^{-1}$ in the upper 120 m than in SOLUB5 (Fig. 6a). In winter, the vertical profile of Fe in ICB-ANNUAL is noticeably different with Fe concentrations higher by ∼0.18 nmol L$^{-1}$ in the upper 50 m and with values that increase and then decrease by ∼0.1 nmol L$^{-1}$ between 50 and 150 m depth whereas Fe concentrations in the SOLUB5

experiment increase gradually in this depth range (Fig. 6b). Thus, when the seasonal variations of iceberg Fe are not considered, the seasonal amplitude of the Fe cycle over the SO is increased with higher concentrations in winter and lower concentrations in summer (Fig. S2a) leading to significant regional differences.



### 3.1.4 Evaluation of the ice shelf contribution

Fe supply from the AIS occurs through two main processes: (1) the basal melting of ice shelves which is coastal, and (2) the calving and melting of icebergs which is more widespread over the SO. Both sources are estimated to be of the same order of magnitude (Depoorter et al., 2013). However, the relative contribution of each source of Fe to the SO Fe pool is not known

due to the lack of data for ice shelves. Here we compared the ISF experiment, which only accounts for the ice shelf Fe source, against the SOLUB5 experiment which encompasses both sources of Fe from the AIS. The surface Fe anomalies in the ISF experiment remarkably differ from the SOLUB5 experiment (Fig. 7). The ice shelf contribution is trapped near the Antarctic coasts extending further offshore in winter (Fig. 7c and 7d) whereas the spatial contribution of icebergs extends the influence of the AIS Fe source significantly more widely over the SO (Fig. 7a and 7b).

The surface Fe concentrations in the ISF experiment are increased by 1 % and 3 % compared to the CTL experiment in summer and winter, respectively. The contribution of ice shelves to the Fe pool over the SO is one order of magnitude lower than in the SOLUB5 experiment which simulates surface Fe concentrations that are increased by 13 % in both seasons. The comparison between both Fe sources suggests two features of the AIS iron source: the high fertilization capacity of icebergs due to a delivery at a longer time scale and the strong limitation that exerts the injection of ice shelf Fe at depth in subsurface

waters deeper than the mixed layer. Moreover, the ice shelf Fe supply occurs in coastal regions that are highly fertilized by sediments leading to elevated Fe concentrations and experiencing already an intense scavenging. Therefore, the additional Fe released from ice shelves is rapidly scavenged and lost from surface waters.

### 3.2 Fertilization effect of the Antarctic ice sheet on surface chlorophyll

The SO is the largest HNLC region where Fe is the main limiting micronutrient for primary productivity. We showed that the

Fe supply by ice shelf and iceberg melting can fertilize the surface waters all year round (Fig. 3). This additional input of Fe can be used at the blooming season by phytoplankton from November to February. Here, we qualitatively evaluate the fertilizing effect of the AIS on surface chlorophyll concentrations (CHL) in summer (December, January, and February). First of all, we briefly compared the CHL climatology from satellite observations of the MODIS-Aqua ocean color product estimated by Johnson et al. (2013) to the CTL experiment (Fig. 8a and 8b). At the scale of the SO, two main qualitative characteristics

can be observed. The CTL experiment represents with a rather good approximation the CHL distribution in summer around the Antarctic continent, from the Antarctic coasts until 65° S. But, in the open ocean, north of 65° S, quite large differences between observations and the standard version of the model are seen especially in the Atlantic sector and in the Pacific sector, north of the Ross Sea.

  To assess the fertilization effect of the AIS on CHL, we computed the CHL difference between the eight experiments and

the CTL experiment (Fig. 9). Globally, the AIS impact on CHL is mostly apparent in the large plume of the Atlantic sector northeast of the Antarctic Peninsula, along the Antarctic coasts in the Indian and Pacific sectors, and, more moderately, in the Pacific sector, north of the Ross Sea. The fertilization effect increases with the Fe solubility with CHL higher by 2 %, 7 %, and 12 % in the SOLUB1, SOLUB5, and SOLUB10 experiments, respectively (Fig. 9a-9c). The mains features driven by the





intensity of the iceberg Fe source are the extension of the Atlantic plume until the Indian sector as well as the increased CHL along the coasts from 80° E until the Ross Sea. In the SOLUB1 experiment, the impact on CHL is particularly low, restricted to the Atlantic sector and in a coastal area around 135° E (Fig. 9a). The plume has the smallest extent from the Antarctic Peninsula until the South Orkney Islands where CHL values are up to ∼0.4 mg m$^{-3}$ higher than in the CTL experiment. The Fe

solubility of 5 % implemented in the SOLUB5 experiment increases significantly the impact of iceberg melting on CHL (Fig. 9b). The Atlantic plume extends eastward, far from the Antarctic Peninsula and the South Orkney Islands. The blooms along the Antarctic coasts in the eastern sector and in the Ross Sea get more intense and two modest plumes emerge north of the Ross Sea and around 90° E. The maximum contribution to CHL between the Antarctic Peninsula and the South Orkney Islands is ∼1 mg m$^{-3}$ higher than in the CTL experiment, and 2.2 mg m$^{-3}$ higher in the coastal area around 135° E. The SOLUB10

experiment exacerbated the spatial patterns described in the SOLUB5 experiment with CHL higher by ∼1.2 mg m$^{-3}$ in the Atlantic sector until the Bouvet Island and up to ∼2.4 mg m$^{-3}$ higher along the coasts in the eastern sector of the SO (Fig. 9c). The plume that extends northward until 60° S from the Ross Sea has more elevated CHL levels, about ∼0.3 mg m$^{-3}$ higher than in the CTL experiment.

In the ICB-SURF experiment, the largest contribution to CHL is simulated with surface chlorophyll concentrations increased

by 12 % over the whole SO. The maximum CHL are by ∼1.3 mg m$^{-3}$ higher in the Atlantic plume, and up to ∼2.5 mg m$^{-3}$ higher in the Ross Sea and along the east coasts relative to the CTL experiment (Fig. 9d). Despite a slightly higher intensity of the bloom, the spatial patterns in the ICB-SURF experiment are very similar to the SOLUB10 experiment (Fig. 9c and 9d). On the contrary, in the ICB-KEEL experiment, the iceberg contribution to CHL is the lowest, surface chlorophyll concentrations being on average only 2 % higher relative to CTL. The Atlantic plume is absent as well as the elevated concentrations along the

Antarctic coasts and in the Ross Sea (Fig. 9f). Nonetheless, even small, a significant fertilizing effect is simulated with CHL values that are locally higher by 0.25 mg m$^{-3}$ than in the CTL experiment. The ICB-ML experiment produces CHL anomalies that lie between the SOLUB1 and the SOLUB5 experiments with maximum CHL up to ∼0.9 and ∼1.6 mg m$^{-3}$ higher than in the CTL experiment in the Atlantic plume and in local areas along the east coasts, respectively (Fig. 9e). The influenced area is clearly smaller than in the SOLUB5 experiment demonstrating that the non-directly available fraction of Fe delivered by

melting icebergs may have a non-negligible impact on CHL at the blooming season. The ICB-ANNUAL experiment simulates CHL levels that are on average higher by 6 % over the SO compared to the CTL experiment with anomalies higher by ∼0.7 mg m$^{-3}$ in the Atlantic plume and up to ∼1.2 mg m$^{-3}$ along the Antarctic coasts (Fig. 9g). Although maximum CHL values in the Atlantic plume are ∼0.3 mg m$^{-3}$ lower, the simulated spatial extent of the CHL anomalies in the Atlantic sector is wider than in the SOLUB5 experiment, the other impacted areas being almost identical in the ICB-ANNUAL and the SOLUB5 experiments.

In the ISF experiment, the increase of CHL is very small as a consequence of the weak impact of ice shelf melting on Fe (Fig. 9h, see subsection 3.1.4).

### 3.3 Model evaluation

The purpose of this sensitivity study is not to specifically improve the skill of the biogeochemical model at representing the Fe and CHL distributions in the SO but to investigate the uncertainties associated to the external source of Fe from the AIS.



However, in order to evaluate that large biases were not introduced by the implementation of the new iron source in the biogeochemical model, we have performed a statistical model-data comparison for Fe and CHL over the SO, south of 50° S. For Fe, we compared the model experiments to a global database constructed by Tagliabue et al. (2012). For surface chlorophyll concentrations, we used a monthly climatology of a satellite-based (MODIS-Aqua) estimates from Johnson et al. (2013). The

statistical comparison shows that performance scores for annual Fe concentrations integrated over the upper 200 m and surface chlorophyll in summer are almost similar in all experiments (Tables S1 and S2). The biases are relatively small ranging to -0.07 to 0.02 nmol L$^{-1}$ for Fe and -0.13 to -0.07 mg m$^{-3}$ for CHL. The main difference is the increase of the mean Fe and surface chlorophyll concentrations showing a better agreement with observations such as in the SOLUB5 experiment for Fe (Table S1) and in the SOLUB10 and ICB-SURF experiments for CHL (Table S2). This statistical analysis reveals no degradation of

the performance skills of the standard version of the biogeochemical model when the external source of Fe from the AIS is added but also no improvements in the spatial distributions of Fe and chlorophyll concentrations. Thus, the absence of glacial Fe fluxes is not a major cause that explains the biased representation of Fe and CHL in the NEMO-PISCES model.

### 3.4  Contribution of the Antarctic ice sheet to primary production and carbon export

The Fe supply from the AIS stimulates, at the blooming season, the phytoplankton activity which can be quantified in terms

of primary production and carbon (C) export. The increase of the annual primary production of phytoplankton (diatoms and nanophytoplankton) integrated over depth is, globally, relatively low in the Fe solubility experiments compared to the total primary production of 2.39 PgC yr$^{-1}$ computed over the SO, south of 50° S, in the CTL experiment (Table 3). The increase in primary production ranges from 0.01 PgC yr$^{-1}$ in the SOLUB1 experiment to 0.12 PgC yr$^{-1}$ in the SOLUB10 experiment, i.e. a difference of one order of magnitude between the least and the most impacted cases. In the SOLUB10 experiment, primary

production is 5 % higher than in the CTL experiment, a difference that drops to less than 1 % in the SOLUB1 experiment. This slightly enhanced primary productivity increases C export by 1 % in the SOLUB1 experiment and by more than 8 % in the SOLUB10 experiment. With a Fe solubility fraction of 5 %, primary production simulated in the SOLUB5 experiment is ∼3 % higher and C export around 5 % higher than in the CTL experiment. Thus, the iron source from the AIS results in a significant but very modest increase in C export at the scale of the SO and subsequent sequestration of carbon in the interior of the ocean.

For the other sensitivity experiments, the predicted impacts on primary production and C export all fall in between those simulated by the SOLUB1 and SOLUB10 experiments. Releasing Fe at the surface as tested in the ICB-SURF experiment produces changes that are only slightly lower than in the SOLUB10 experiment. This suggests that the efficiency of the AIS source is higher when located at the surface. The comparison of the SOLUB5 experiment with the ICB-ML experiment reveals that the non-directly available fraction of Fe released from icebergs may increase by ∼40 % the impact of the source on

primary production and C export. The ICB-ANNUAL experiment shows a primary production and a C export almost equals to the SOLUB5 experiment suggesting no effect of the seasonal variability of iceberg Fe supplies on annual primary productivity and C export at the scale of the SO. Finally, when only the ice shelf source of Fe is considered in the ISF experiment, primary production and C export are almost unchanged.





## 4 Discussion

### 4.1 Sensitivity of Fe and chlorophyll to the iron source from the Antarctic ice sheet

Our sensitivity study aims at delineating the biogeochemical impacts of the uncertainties surrounding the fertilization capacity of the AIS. Different aspects of the AIS Fe fluxes have been explored: the intensity of the source, the impact of the iceberg Fe distribution along the water column, and the contribution of the seasonal variations of the iceberg meltwater.

The Fe supply from the AIS is highly sensitive to the hypothesized solubility of ferrihydrite revealing strong impacts on the spatial distribution of Fe. The main supply of Fe occurs in the Atlantic sector downstream of the Antarctic Peninsula, along the Antarctic coasts, and, more moderately, in the Ross Sea. The iceberg contribution to surface Fe is large and can extend until 50° S as shown by the large plume expanding from the Antarctic Peninsula until the Indian sector (Fig. 3). The spatial distribution of the surface Fe anomalies in our model setup is in line with Laufkötter et al. (2018) but differs substantially from Death et al. (2014). In Death et al. (2014), the main fertilized area is simulated along the eastern sector of the Antarctic coasts showing a larger offshore extent, the Atlantic plume is clearly much less marked and extended, and the AIS influence in the region of the Ross Sea is weaker. These differences may be linked to the implementation in Death et al. (2014) of a basal iceberg sediment loading which induces high Fe concentrations in the basal layer and very low concentrations above this basal layer whereas an homogeneous distribution is considered in our study. Their vertically-varying distribution of Fe in icebergs may simulate a stronger fertilization effect in the calving regions driven by an important basal melting. Further offshore, once the basal Fe rich part of icebergs has melted, the release of Fe is strongly decreased due to the lower Fe concentrations in the upper part of icebergs, resulting in a weaker fertilization effect in remote areas of the open ocean such as in the Atlantic sector.

The AIS fertilization impact on surface chlorophyll depends on the intensity of the AIS Fe source as well as on the choice of its vertical distribution (Fig. 9). The efficiency of the fertilization is regionally important with increased CHL along the east coasts of Antarctica and in the core of the Atlantic plume off the tip of the Antarctic Peninsula. However, at the scale of the SO, south of 50° S, the AIS impact on primary production is quite modest reaching a maximum increase of 5 % in our set of experiments relative to the control run (Table 3). Our results are similar (lower by 3 %) to Wadley et al. (2014) but contrast sharply with the 30 % increase in primary production estimated in Death et al. (2014). The AIS contribution in Death et al. (2014) is evaluated against atmospheric dust, sediments being not taken into account. The lack of the sedimentary Fe source, estimated to be the largest in the SO (Borrione et al., 2014; Lancelot et al., 2009; Tagliabue et al., 2009, 2014a; Wadley et al., 2014), leads to increase significantly the fertilization effect of icebergs and ice shelves, particularly in coastal regions where sediment supplies have a large influence. Our study suggests that the AIS fertilization effect is weaker than suggested by Death et al. (2014), especially in coastal areas, as a consequence of the large input of Fe from sediment remobilization.

The enhanced primary production increases the C export by 8.4 % in our most impacted case (Table 3), a result significantly lower than the increase in particle export of 30 % in Laufkötter et al. (2018). The reasons for such a difference are difficult to disentangle as the modeled Fe fluxes from the AIS and sediments are in the same order of magnitude between both studies. A potential difference in both modeling setups may arise from a different treatment of sediment mobilization, in particular in the description of the horizontal and vertical distribution of sediments. However, while low at the scale of the SO, the fertilization





effect of the AIS on primary productivity and C export can be regionally significant as pointed out by observations (Duprat et al., 2016; Herraiz-Borreguero et al., 2016; Smith et al., 2007; Wu and Hou, 2017) For instance, in the highly fertilized area of the Atlantic plume, northeast of the Antarctic Peninsula (36° W-56° W, 58° S-63° S), primary production and C export are increased by ∼30 % and by ∼42 %, respectively, in the SOLUB10 experiment compared to the CTL experiment (Table 3), i.e.

5 to 6 times higher than at the scale of the whole SO.

Climatically our study points out that the fertilization effect of the AIS on C export is moderate on time scales of 50 to 100 years. However, when integrated over time scales of thousands years, the role played by the AIS on the carbon sequestration might be important and be evaluated as a key component such as atmospheric dust iron for glacial-interglacial regulation of the carbon cycle (Martin, 1990). In a climate change perspective, our results suggest that any change in the supply of Fe from an

increased melting of icebergs and ice shelves should result in a quite moderate impact on ocean biogeochemistry and export production at the scale of the whole SO. Indeed, doubling the AIS Fe fluxes in the SOLUB10 experiment increases by only ∼3.6 % the C export compared to the SOLUB5 experiment (Table 3). Nevertheless, at a more local scale, the fertilization effect of the AIS induced by global warming could be drastically strengthened with potentially important consequences for phytoplankton physiology, nutrient availability and marine ecosystems (Boyd, 2019; Boyd et al., 2010, 2015; Hopwood et al.,

15    2017).

The choice of the iceberg Fe source distribution leads to significant differences in the magnitude of the fertilization effect. In the case of a surface distribution, the effect is maximum. All the Fe delivered by the iceberg meltwater flux to the mixed layer is available to sustain primary productivity in spring and summer and strongly affects the vertical profiles of Fe particularly in highly fertilized areas (Fig. 6). This theoretical distribution may lead to an overestimated supply of Fe in summer when the

mixed layer is highly stratified, particularly in the case of large icebergs, partially ignoring the specific role of the Fe delivered below the MLD. Antarctic icebergs have different shapes (Romanov et al., 2012) and class-size categories (Silva et al., 2006; Tournadre et al., 2015, 2016) both evolving during their life cycle (Bouhier et al., 2018). Moreover, the sediment distribution within icebergs is highly heterogeneous (Hopwood et al., 2017; Raiswell et al., 2016). All these features combined with distinct regimes of iceberg melting (FitzMaurice et al., 2017) fully constrain the delivery of Fe along the water column and below the

mixed layer. Thus, the inherently heterogeneous nature of icebergs and its temporal evolution is extremely difficult to consider and to implement in a model. The choice of a surface distribution might be inappropriate to represent the iceberg supply in the ocean but without any degree of certainty. In fact, measured vertical profiles of Fe concentrations around icebergs in the Bellingshausen Sea in summer (De Jong et al., 2015) and in the Weddell Sea in autumn (Lin et al., 2011) suggest that both ICB-SURF and SOLUB5 experiments simulate vertical distribution of Fe that could be observed in the wake of melting icebergs. At

least, based on future observations, the representation of the iceberg Fe source could be better constrained and parameterized in models.

While the iceberg freshwater fluxes vary monthly (Fig. 1a), the AIS contribution to the Fe pool is almost equally effective in summer and winter, mainly driven by the balance between high AIS Fe fluxes and phytoplankton consumption in summer, and low AIS Fe fluxes and light limitation in winter. However, the seasonal variations of the iceberg Fe fluxes contribute to

significant differences in the spatial distribution of Fe (Fig. 4g and 4h) which have small impacts on annual primary production





and C export when integrated over the SO (Table 3). The spatial differences in surface chlorophyll are globally modest in summer between the SOLUB5 and the ANNUAL experiments. Nevertheless, the larger amplitude of the Fe cycle over the SO in the ANNUAL experiment (Fig. S2a) modulates the seasonality of surface chlorophyll during the growing season: the bloom initialization occurs earlier, the bloom apex in December is higher and the bloom decay is faster from January to April (Fig.

S2b). Thus, the monthly variations of the iceberg Fe supply alters the seasonal cycle of chlorophyll in the SO.

## 4.2    Model caveats and uncertainties

A surprising result that may be linked to a potential model deficiency is the absence of iron fertilization effect in the very close vicinity of the Antarctic coasts. This can be observed in the difference of CHL between the SOLUB5 experiment and the CTL experiment (Fig. 9b). In fact, none of the iceberg fertilization experiments shows an increase in chlorophyll near the

Antarctic shores (Fig. 9). This unexpected result is due to a strong and systematic nutrient limitation in summer simulated by the biogeochemical model (Fig. S1). The seasonal cycles of nutrients at a station near the shore of the Amundsen Sea (106° W, 75° S) in the CTL and SOLUB5 experiments display a marked limitation in NO3, PO4, and Si in January and February (Fig. S1a-S1c) whereas Fe is non limiting (Fig. S1d). The nutrient limitation strongly affects CHL in both experiments, the seasonal cycles being almost similar (Fig. S1e). The nutrient limitation may occur locally along the Antarctic coasts, however high

levels of primary productivity in spring and summer are observed in large regions such as in the numerous coastal polynyas present in the SO (Arrigo and van Dijken, 2003; Arrigo et al., 2015). Thus, this possible biased behavior of our model may result from missing processes or sources that may supply macro-nutrients in the mixed layer such as the oceanic circulation in ice shelf cavities (Herraiz-Borreguero et al., 2015; Jacobs et al., 2011; White et al., 2019), the glacial meltwater runoff (Beaton et al., 2017; Hawkings et al., 2015, 2017; Hodson et al., 2017) or the melting of ice shelf and ice sheet (Arrigo et al., 2017,

2015; Hawkings et al., 2015; Pritchard et al., 2012; St-Laurent et al., 2017; Wadham et al., 2016). Another process that can be advocated is the entrainment of nutrient-rich waters induced by basal melting of grounded glaciers, a physical mixing process observed for Greenland glaciers which highlights the role of subglacial discharge plumes on upwelling of macro-nutrients such as $NO_3$ in the euphotic zone (Hopwood, 2018; Kanna et al., 2018; Meire et al., 2017).

We highlighted that the distribution of the iceberg Fe fluxes below the MLD may represent a non-negligible fraction of

bioavailable Fe for primary productivity. Indeed, the iceberg Fe delivery at depth in the SOLUB5 and ICB-KEEL experiments feed a subsurface reservoir of Fe that can supply surface waters by the deepening of the MLD through subseasonal storms (Nicholson et al., 2016; Swart et al., 2015) or deep mixing (Tagliabue et al., 2014b). We suggest that this distribution of the iceberg Fe source has to be considered if implemented in biogeochemical models. However, in our sensitivity study, we only applied one average depth of the submerged part of icebergs whereas several class sizes coexist in the SO where large tabular

to small icebergs are observed covering a size range of 0.1-10000km$^2$ (Silva et al., 2006; Tournadre et al., 2016, 2015). The size evolution of icebergs along their life cycle is poorly documented, but fragmentation is a significant mechanism process in the reduction of their size which increases the iceberg melt (Bouhier et al., 2018). This process impacts the time variations of the delivery of bioavailable Fe at depth that we did not explored here. Moreover, the distribution of the iceberg Fe fluxes along

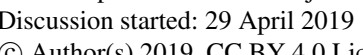



the water column, i.e. around and below icebergs, is probably not homogeneous as reported in De Jong et al. (2015) and Lin et al. (2011) giving an additional uncertainty not explored in this study.

A large uncertainty in the fertilization capacity of Fe delivered by icebergs and ice shelves comes from the intrinsic nature of this sedimentary source. Indeed, a very large fraction of Fe found in icebergs has a lithogenic origin (Raiswell et al., 2006; Shaw et al., 2011). The supply of lithogenic Fe can be separated into three categories: the labile Fe directly bioavailable, the semi-labile particulate Fe that will not dissolve rapidly once released to seawater, and the refractory insoluble fraction. We focused our study on the first fraction. However, the semi-labile fraction may have a significant contribution in fertilizing the surface waters of the SO. If lithogenic Fe with a low dissolution rate is not scavenged or experiences low sinking speeds (nanoparticles), this fraction can be maintained in the upper layer and be bioavailable at long time scales. This residence time may strongly affect the dissolved iron distribution from icebergs over the SO. As particulate lithogenic Fe is a significant pool of Fe in icebergs (Raiswell et al., 2006; Raiswell, 2011; Shaw et al., 2011), the contribution of the non-directly bioavailable fraction to surface dissolved iron can be higher than actually observed. However, nothing is known about the fraction of lithogenic Fe bioavailable at long time scales as well as on its quantity.

## 5 Conclusions

We implemented in the biogeochemical model NEMO-PISCES (Aumont et al., 2015) the external source of iron from the Antarctic Ice Sheet based on recent estimations of Antarctic meltwater fluxes from icebergs and ice shelves (Depoorter et al., 2013; Merino et al., 2016). The modeled Fe fluxes from the AIS are in the range of previous modeling studies (Death et al., 2014; Laufkötter et al., 2018) and in the lower range of recent estimates from data (Raiswell et al., 2016). We investigated the impacts of different sources of uncertainties related to the AIS iron source on Fe and surface chlorophyll distributions: the solubility of Fe, the vertical distribution of the iceberg source and its seasonal variability. Large differences in the AIS iron fertilization are ultimately attributable to varying Fe solubility (1-10 %), currently poorly constrained by observations (Boyd et al., 2012). The supply of Fe from the AIS is significant in the Atlantic sector northeast of the Antarctic Peninsula and along the Antarctic coasts, particularly in the eastern sector, with large implications for the magnitude of phytoplankton blooms. The surface Fe and chlorophyll concentrations are increased by 3 to 25 % and by 2 to 12 %, respectively, at the scale of the SO. The contribution of Fe released from ice shelves is restricted to coastal areas with very small impacts on chlorophyll and primary productivity whereas modeled Fe fluxes from ice shelves and icebergs are almost similar. Our results also underline the role played by the vertical distribution of the iceberg Fe source due to the potentially non-negligible contribution of Fe delivered below the MLD. This non-directly available supply can not be considered as a lost fraction for primary production but as a subsurface reservoir. The variability of the AIS contribution to the SO Fe pool is strongly linked to the interplay between the seasonal variations of meltwater released from icebergs and the physical and biological processes that characterize the dynamic of the SO: light limitation, MLD variations, iron limitation, and Fe consumption by phytoplankton. At the scale of the SO, the AIS fertilizing effect on primary production, mainly driven by icebergs, is relatively weak but with a non-negligible contribution to C export: primary production and C export are increased by 5 % and 8.4 %, respectively, in the most contributive



case compared to our control experiment. However, in highly fertilized regions in the Atlantic sector and along the Antarctic coasts, the AIS impact is significant, primary production and C export being increased by up to 30 % and 42 %, respectively. Our results over the SO are noticeably lower than the AIS Fe contribution to the marine particle export recently estimated to 30 % in Laufkötter et al. (2018). This significant difference reveals the necessity to pursue *in situ* observations, particularly
to better constrain the distribution of Fe in the water column in the close vicinity of icebergs, as well as modeling studies to reduce the large uncertainties that encompass the AIS source of Fe. Indeed, representing the biogeochemical features of the SO in ocean models is particularly challenging, however we argue that the integration of the AIS iron source may help to fill the gap of misrepresented characteristics in the SO and to represent the complex cycle of Fe in the SO (Boyd and Ellwood, 2010). Moreover, since the Antarctic continental ice sheet has experienced a significant reduction of its mass (The IMBIE team, 2018)
that may continue and amplify in the near future due to climate change (Rintoul et al., 2018), it could be particularly relevant to integrate the AIS Fe source in biogeochemical and climate models in order assess its role to marine ecosystems and take into account potential negative feedbacks on climate change (Barnes et al., 2018). However, according to the modest impacts we find in our study we can speculate a relatively moderate increase of primary production and C export to climate change until the end of the present century in the SO.

*Code and data availability.* The version code of the NEMO model, including PISCES-v2, used for this study is freely available at https: //www.nemo-ocean.eu/. To access the NEMO svn repository, users should register on the NEMO website at https://forge.ipsl.jussieu.fr/ nemo/register. Model data are available at https://doi.org/10.5281/zenodo.2633097

*Author contributions.* The author contributions to this paper are as follows.
– Conceptualization: RP and OA
– Formal analysis: RP
– Funding acquisition: OA and LB
– Investigation: all
– Methodology: RP, OA, and GM
– Validation: RP and OA
– Visualization: RP
– Writing original first draft: RP
– Writing, review: all

*Competing interests.* The authors declare that they have no conflict of interest.





*Acknowledgements.* We would like to thank the ANR SOBUMS (ANR-16-CE01-0014) and EU-H2020-CRESCENDO (grant agreement no. 641816) projects for funding. This work was performed using HPC resources from GENCI-IDRIS (Grant 2018-A0040107451). The corrected ocean color product was retrieved from Australia's Integrated Marine Observing System (http://imos.org.au/). The dissolved iron database is available on the GEOTRACES International Data Assemble Centre (https://www.bodc.ac.uk/geotraces/data/historical/).



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




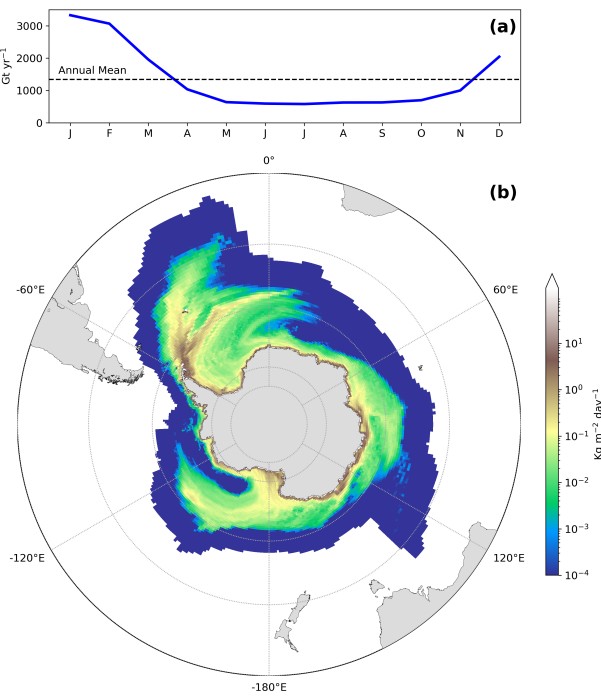

**Figure 1.** (a) Seasonal cycle of iceberg freshwater fluxes over the Southern Ocean from the climatology of Merino et al. (2016). (b) Annual mean freshwater fluxes from icebergs (Merino et al., 2016) and ice shelves (Depoorter et al., 2013) over the Southern Ocean, south of 30° S, used to represent the Fe supply from the Antarctic ice sheet in our study. Note the logarithmic scale.





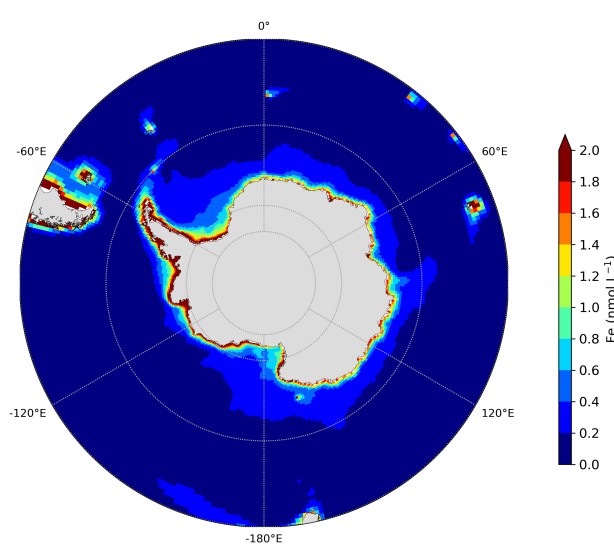

**Figure 2.** Annual mean of surface Fe concentrations in the Southern Ocean, south of 45° S, in the CTL experiment.




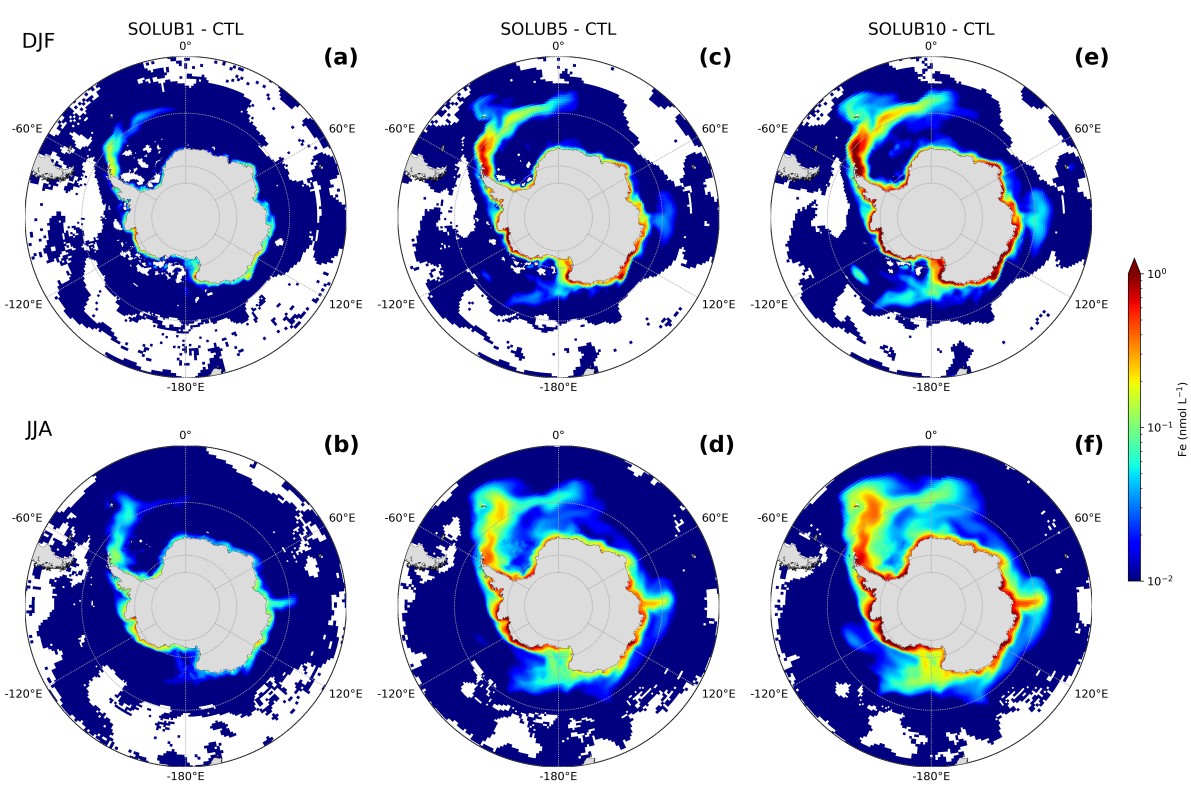

**Figure 3.** Difference in surface Fe concentrations between the (a and b) SOLUB1, (c and d) SOLUB5, (e and f) SOLUB10 experiments and the CTL experiment (experiments minus CTL) in (a, c, and e, upper row) summer (December, January, and February) and (b, d, and f, lower row) winter (June, July, and August) in the Southern Ocean, south of 45° S. Note the logarithmic scale.





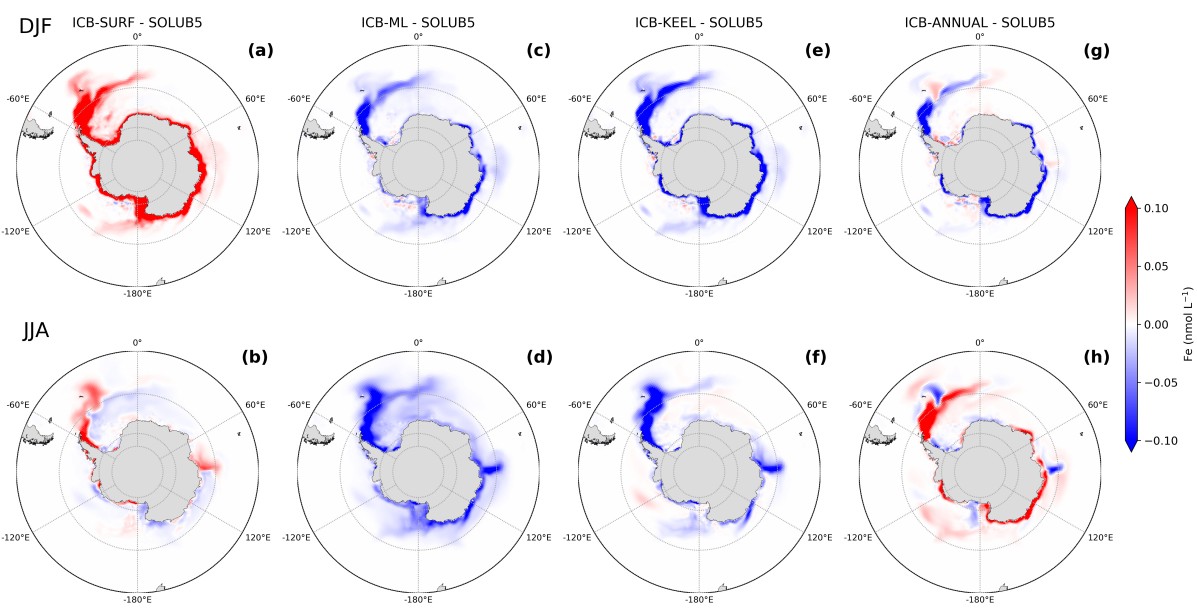

**Figure 4.** Difference in surface Fe concentrations between the (a and b) ICB-SURF, (c and d) ICB-ML, (e and f) ICB-KEEL, (g and h) ICB-ANNUAL experiments and the SOLUB5 experiment (experiments minus SOLUB5) in (a, c, e, and g, upper row) summer (December, January, and February) and (b, d, f, and h, lower row) winter (June, July, and August) in the Southern Ocean, south of 45° S.





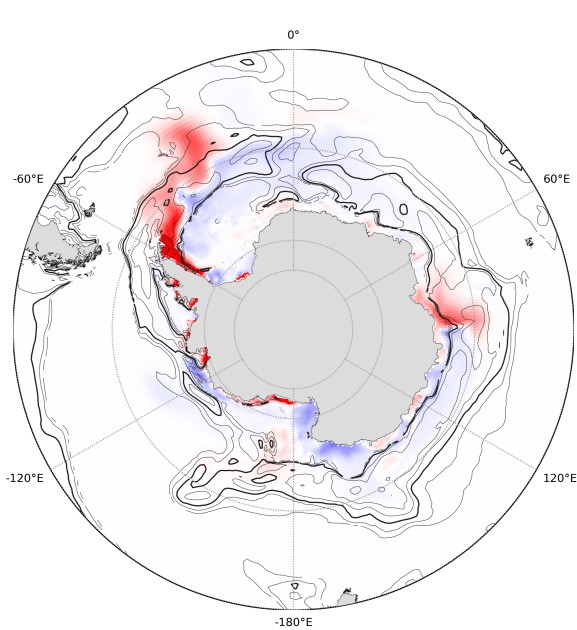

**Figure 5.** Difference in surface Fe concentrations between the ICB-SURF and the SOLUB5 experiments in winter (June, July, and August) in the Southern Ocean, south of 45° S. The black isoline represents the mixed layer depth at 120 m and the grey isolines represent mixed layer depth shallower than 120 m.



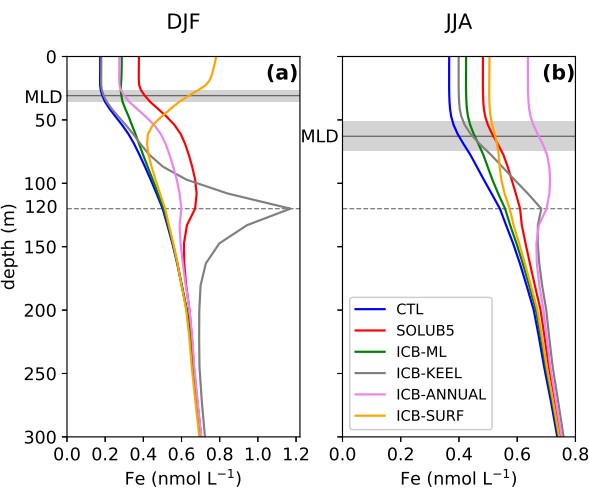

**Figure 6.** Vertical profiles of Fe concentrations until 300 m depth northeast of the Antarctic Peninsula (36° W-56° W, 58° S-63° S) in the CTL (blue), SOLUB5 (red), ICB-ML (green), ICB-KEEL (grey), ICB-ANNUAL (pink), and ICB-SURF (orange) experiments in (a) summer (December, January, and February) and in (b) winter (June, July, and August). Solid light grey line is the seasonal mean depth of the MLD in (a) summer and (b) winter, in grey shading is the standard deviation of the MLD over the region in (a) summer and (b) winter, and the dashed gray line is the 120 m depth.



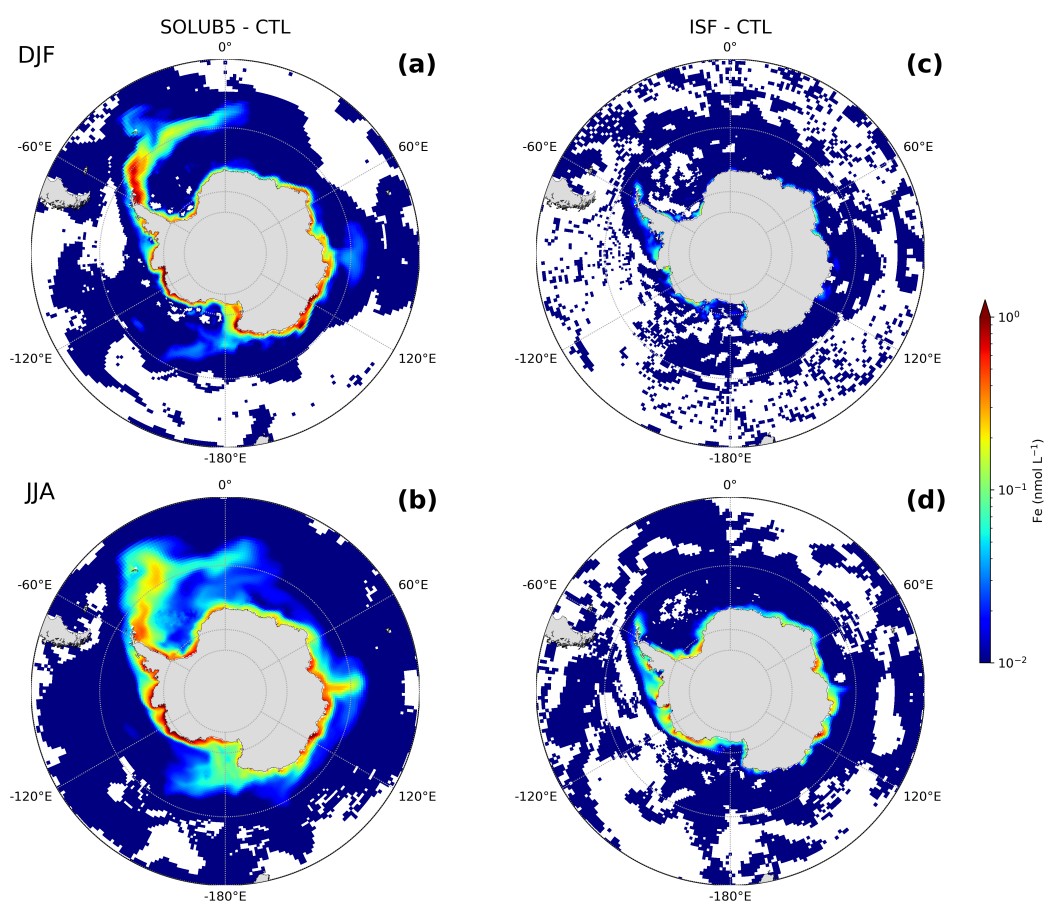

**Figure 7.** Difference in surface Fe concentrations between the (a and b) SOLUB5, (c and d) ISF experiments and the CTL experiment (experiments minus CTL) in (a, and c, upper row) summer (December, January, and February) and (b, and d, lower row) winter (June, July, and August) in the Southern Ocean, south of 45° S. Note the logarithmic scale.





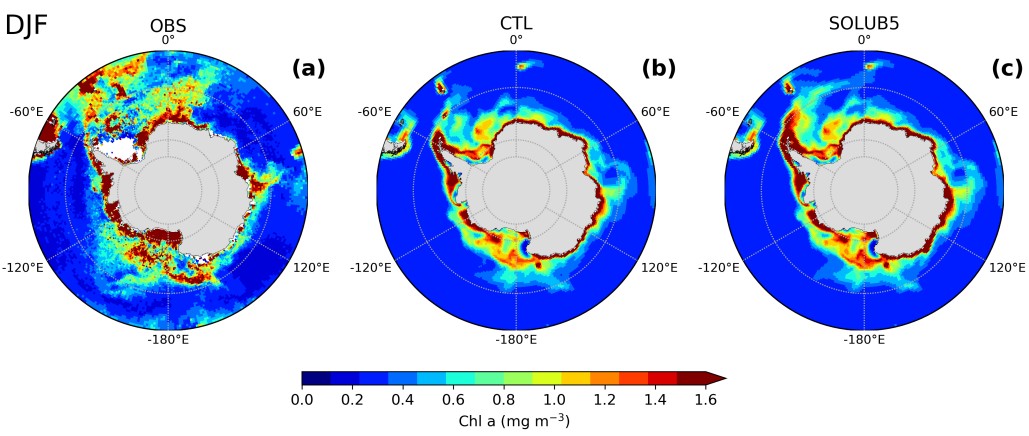

**Figure 8.** Difference in surface Fe concentrations between the (a and b) SOLUB5, (c and d) ISF experiments and the CTL experiment (experiments minus CTL) in (a, and c, upper row) summer (December, January, and February) and (b, and d, lower row) winter (June, July, and August) in the Southern Ocean, south of 45° S. Note the logarithmic scale.





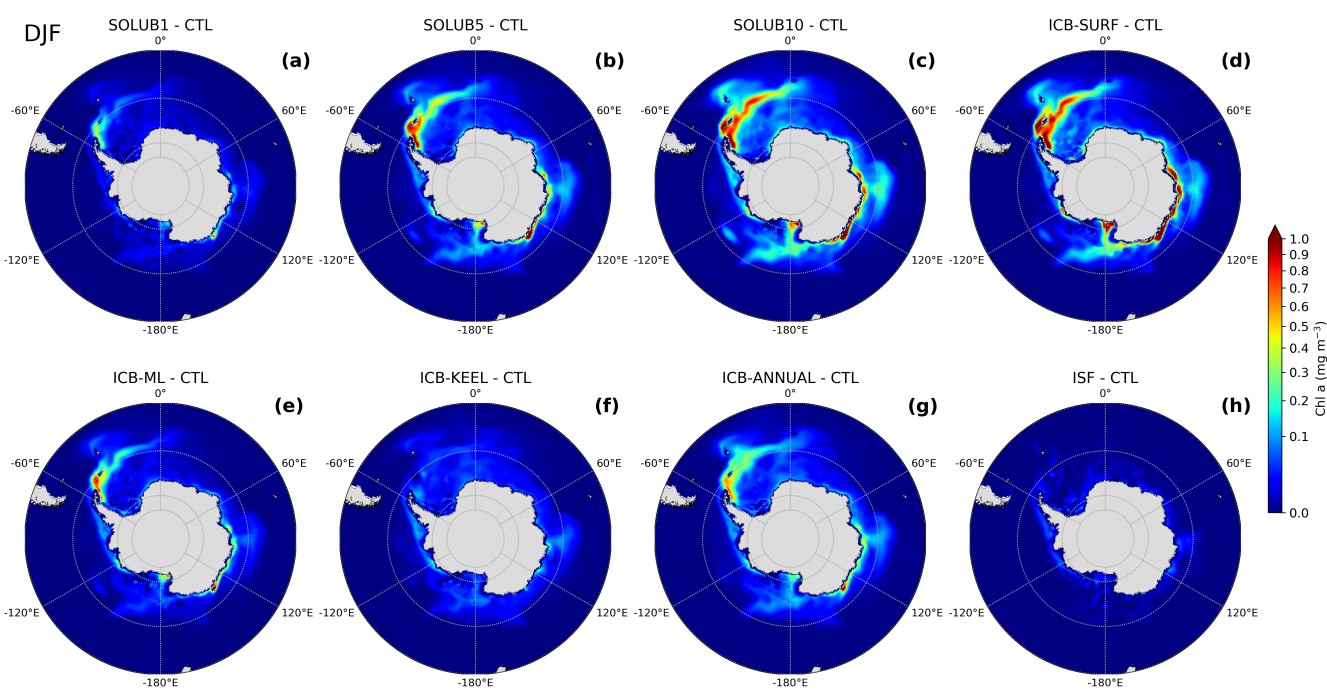

**Figure 9.** Difference in surface chlorophyll concentrations in summer (December, January, and February) between the (a) SOLUB1, (b) SOLUB5, (c) SOLUB10, (d) ICB-SURF, (e) ICB-ML, (f) ICB-KEEL, (g) ICB-ANNUAL, (h) ISF experiments and the CTL experiment (experiments minus CTL) in the Southern Ocean, south of 45° S.



**Table 1.** Annual estimates of bioavailable Fe fluxes from observational and modeling studies in the SO. The iceberg bioavailable Fe flux from Raiswell et al. (2016) is calculated applying a Fe solubility of 10 % to their estimates of potentially bioavailable Fe fluxes.

| References | Fe Flux (Gmoles $yr^{-1}$) | | |
|---|---|---|---|
| | Iceberg | Ice shelf | Total |
| Raiswell et al. (2008) | 1.07 – 2.15 | - | - |
| Raiswell et al. (2016) | 0.32 – 2.5 | - | - |
| Shaw et al. (2011) | 0.72 – 7.2 | - | - |
| Hawkings et al. (2014) | - | 1.1 – 3 | - |
| Death et al. (2014) | 1.16 | 0.16 – 1.6 | 1.32 – 1.76 |
| Laufkötter et al. (2019) | 0.05 – 2.54 | 0.06 – 0.6 | 0.11 – 3.14 |
| Our study | 0.12 – 1.2 | 0.13 – 1.3 | 0.25 – 2.5 |





**Table 2.** Description of model experiments. The ice shelf Fe release distribution is applied between the base and the grounding line of ice shelves following the parameterization of Mathiot et al. (2013). The climatology of ice shelf Fe fluxes is annual.

| References | Iceberg source | Ice shelf source | Fe solubility (%) | Iceberg Fe release distribution | Climatology of Iceberg Fe Fluxes |
|---|---|---|---|---|---|
| CTL | no | no | 0 | n.a. | monthly |
| ISF | no | yes | 5 | n.a. | n.a. |
| SOLUB1 | yes | yes | 1 | 0 - 120 m | monthly |
| SOLUB5 | yes | yes | 5 | 0 - 120 m | monthly |
| SOLUB10 | yes | yes | 10 | 0 - 120 m | monthly |
| ICB-SURF | yes | yes | 5 | surface | monthly |
| ICB-ML | yes | yes | 5 in ML - 0 below ML | 0 - 120 m | monthly |
| ICB-KEEL | yes | yes | 5 | at $\sim$120 m | monthly |
| ICB-ANNUAL | yes | yes | 5 | 0 - 120 m | annual |

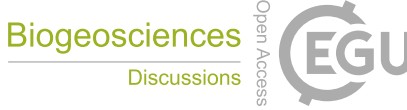



**Table 3.** Annual primary production integrated over depth (PP) and C export at 150 m depth in the CTL experiment and in the AIS Fe source experiments over the Southern Ocean, south of 50° S. In brackets are the increase of PP and C export relative to the CTL experiment in the highly fertilized plume of the Atlantic sector, northeast of the Antarctic Peninsula (36° W-56° W, 58° S-63° S).

| References | PP (PgC yr$^{-1}$) | % increase PPL from CTL | C export 150 m (PgC yr$^{-1}$) | % increase C export from CTL |
|---|---|---|---|---|
| CTL | 2.39 | | 0.63 | |
| ISF | 2.39 | 0.1 | 0.63 | 0.3 |
| SOLUB1 | 2.40 | 0.7 (7) | 0.64 | 1.1 (8) |
| SOLUB5 | 2.46 | 2.9 (24) | 0.66 | 4.8 (30) |
| SOLUB10 | 2.51 | 5.0 (32) | 0.68 | 8.4 (42) |
| ICB-SURF | 2.49 | 4.3 (35) | 0.68 | 7.5 (45) |
| ICB-ML | 2.43 | 1.6 (15) | 0.65 | 2.6 (20) |
| ICB-KEEL | 2.42 | 1.2 (2) | 0.64 | 2.1 (3) |
| ICB-ANNUAL | 2.45 | 2.8 (21) | 0.66 | 4.7 (28) |