# Peer review of "Sensitivity of ocean biogeochemistry to the iron supply from the Antarctic Ice Sheet explored with a biogeochemical model"

_Biogeosciences, 2019_

## Referee Comment (RC1) · Anonymous Referee #1 · 22 May 2019

Review 'Sensitivity of ocean biogeochemistry to the iron supply from the Antarctic ice sheet explored with a biogeochemical model'

The authors present a model study investigating how Fe input into the Southern Ocean from icebergs and the Antarctic Ice Sheet affects the distribution of Fe and primary production in the marine environment. Recognizing the uncertainty in the magnitude and nature of these Fe sources, and thus several difficulties in meaningfully parametrizing them to date, the authors opt to model several scenarios with important differences in Fe solubility and the distribution of melt-derived Fe in the water column. The results, with respect to primary production and C export, fall within the (very broad) range of

other model studies suggesting a modest impact of this Fe on Southern Ocean productivity. A key strength of this specific study is that it makes considerable effort to highlight the many uncertainties surrounding this Fe source. Numerous other recent works have proposed much stronger effects but neglected to consider some, or all, of the uncertainties highlighted herein. Whilst there are a few areas in the text where I think some improvements can be made, I generally therefore consider this to be a valuable addition to the field, suitable for publication in BGS and, in my opinion, one of the most comprehensive manuscripts on the subject of modeling these Fe fluxes to date.

My expertise is in biogeochemistry, I defer to a more qualified reviewer for issues concerning details of the model used. Before returning the text to the journal, it would benefit slightly from a read through from an English editor.

General comment; have the authors considered the meltwater 'pump' effect outlined in some recent work (see comment on page 4, (Cape et al., 2019; St-Laurent et al., 2017, 2019)? I wasn't clear if this effect would be captured in the model or not.

General comment: How is C export scaled to primary production in the model, does the model successfully replicate the observed relationship between the two? Looking at some other models and calculations in the literature, it appears to me that a key reason why very broad ranges are often quoted for C export from specific Fe fertilization scenarios is simply because of the way Fe or productivity/chlorophyll a is scaled to C export. The 'high' C export estimate of (Duprat et al., 2016) is scaled linearly with chlorophyll/productivity –which is not consistent with observational Southern Ocean data. It is not clear to me if this is also a problem with the (Laufkötter et al., 2018) model which matches the Duprat calculation surprisingly well producing a fertilizing effect significantly above that found herein. (Observations with multiple methods show that C export efficiency declines sharply with increasing productivity in the Southern Ocean, although the precise reason(s) for this seem to be unclear (Maiti et al., 2013; Le Moigne et al., 2016)).

[Figure]

Specific comments by Page/line Title: Antarctic Ice Sheet 1/12 'seasonal variations' in the timing of melting? If I understand correctly, this sentence would read better 'Seasonal variations make almost negligible differences...'

2/3 Raiswell 2016 does not contain extensive atmospheric dust work, I am sure there are better values/references for dust deposition

2/14 'the mean flux'. You mean the total flux?

2/16 'the few modelling studies conducted to date'

2/27 'fueling surface waters'. You mean 'fueling' productivity or just delivering Fe?

2/28 Not sure this is accurate, it has been speculated that glacially derived Fe was fueling primary production in the Southern Ocean for some time e.g. (Hart, 1934), it just has proved very difficult to quantify.

2/32 See also (Wu and Hou, 2017) - a particularly interesting read as it, when compared to (Duprat et al., 2016) demonstrates that there are significant differences in observational data constraining the effect of icebergs, not just in the models.

2/34 'the Prydz bay'. Delete the

3/12 . . .will increase the supply of Fe. . . Assuming that the Fe input scales linearly with ice-melt, which may be a little speculative

3/18 'along the water column' means horizontal, you mean 'through'

4/10 Here something concerning the 'meltwater pump' may be relevant. High Fe concentrations adjacent to Icesheets (in the ocean) would generally be attributed to direct input from melt/sediment release etc, but release of meltwater can also 'pump' ambient to the surface and thus bring Fe from shelf sediments and the sub-surface Fe reservoir into surface waters. These effects are difficult to tease-apart from field data. But some model calculations suggest that the magnitudes of Fe from 'pumping' and from direct input (melt/freshwater/freshwater derived particles) are comparable – all be it with large

uncertainties. See (Cape et al., 2019; St-Laurent et al., 2017, 2019) for overviews of this effect and what we do/don't know about it.

4/23 It is not clear to me what the % here refer to, I guess the % weight of sediment which is ferrihydrite, but please clarify (also specifically what is the mean – a global mean??)

4/24 This is a muddled concept in the field in general. All labile Fe could be potentially 'biologically available' if processed/delivered in the right way, I would stick specifically with the 'soluble' fraction rather than trying to define a 'biologically available' fraction as this is an arbitrary exercise. The concept of 'utilization' (Boyd et al., 2012) is perhaps more useful as 'bioavailability' is a qualitative term.

4/30 Seems like an odd thing to say. 'no data allow the constraining of...' or 'allow us to'

5/30 The 'buoyancy effect' is widely attributed with bringing iceberg-derived components (e.g. particles/Fe) to the surface, but as far as I'm aware there isn't much clear evidence of it actually doing this, or even much data to show how ice melt behaves in the real world. An alternative argument is that something akin to convective cells develop up the sides of the iceberg, and that these reach neutral buoyancy before they reach the surface i.e. most melt doesn't 'rise' to the surface. In any case, there is certainly very limited data to show how ice melt behaves around icebergs (Helly et al., 2011; Stephenson et al., 2011).

6/30 How do these concentrations compare to 'real' Fe concentrations in these areas?

7/9&10 This line 'Furthermore, in winter,...' does not make sense

7/6 These concentrations are not feasible, how is scavenging constrained? Such a high dissolved Fe concentration (27 nM) would, practically immediately, precipitate.

3.1.4 Whilst the effect is poorly defined, the meltwater 'pump' should be at least mentioned here.(St-Laurent et al., 2017, 2019)

10/33 The mains

11/11 the Bouvet Island. Delete 'the'

11/20 Nevertheless, though small

11/25 CHL at the blooming season, you mean 'throughout' or 'during the season' (general comment CHL is, at a glance, similar to CTL, maybe use 'Chl a' or similar)

12/30 equal to

12/33 'are almost unchanged.' Compared to?

13/27 'leads to a significant increase in'

13/32 Indeed. The first thing I did after reading this study was to refer to (Laufkötter et al., 2018). I was very surprised to find that both studies use very similar parameterizations for the total Fe input. As a biogeochemist, my simplistic conclusion is therefore that these results (collectively) are not reproducible between models, as completely opposite conclusions are reached using practically the same Fe input. More surprising is that the results of the studies don't even overlap- given that both studies use very broad ranges in Fe input which were designed to span all environmentally relevant scenarios. This is problematic, because it makes the studies (again, collectively-this is not a specific critique of this study) impossible to interpret from a biogeochemical perspective. So the critical question is why is there such a large difference? The authors herein do a generally good job of discussing the differences between existing iceberg models, but perhaps this information (presently in the text) could be thinned a little and compiled in the form of a table which would at least eliminate some causes of differences between independent models. As a biogeochemist it is difficult to comment further other than to raise a flag that model results should be treated with extreme caution until some consensus can be found between different model studies.

14/2 Yes, but be careful here concerning 'regionally sig. C export'. Compare (Wu and Hou, 2017) and (Schwarz and Schodlok, 2009) with (Duprat et al., 2016), the later

study claims a much larger effect, but only in the C export calculated, I suspect this is largely because of how the observed data (chlorophyll) is scaled to C export and thus reflects different assumptions in the calculation rather than actual differences in the raw data.

14/29 Does (De Jong et al., 2015) not conclude that much of the Fe is sub-surface?

15/19 'runoff' [as a macronutrient source] this is a bit of a misleading statement, even in the North Atlantic, where macronutrient concentrations are much lower in the mixed layer, runoff dilutes the concentration of N and P macronutrients (Meire et al., 2016), so a missing macronutrient-runoff source couldn't plausibly explain the problem herein. Similarly ice contains very low macronutrient concentrations.

15/22 This seems more plaussible, see for example (Cape et al., 2019), although even these 'upwelled' nutrient fluxes would be modest and I doubt sufficient to explain the model problem-plus they would come with Fe. In these references here, I think the authors mean (Hopwood et al., 2018) rather than the Hopwood paper listed. Alternatively, how scavenging is accounted for in the model (a difficult thing to do) presumably could cause this effect, if Fe is removed a little too slowly, it will 'over-fertilize' in the model world and thus, all other things being equal, drawdown macronutrients much faster than would be the case otherwise. As noted, I am not a model expert, but I would guess that macronutrient distributions in the model match real data better than Fe distributions and thus would speculate that problems are more likely to arise from how Fe is parametrized than with macronutrient sources/sinks.

16/2 See also (Boyd et al., 2012) – specifically the 'utilization' of Fe shifts significantly along 'Iceberg Alley'

16/7 Perhaps, but then this becomes a question of organic ligands and to what extent these are able to transfer Fe into the dissolved phase. I'm not aware of any work around icebergs looking at ligand-iceberg interactions, but this has been investigated with respect to glacially derived particles, for general discussion of how ligands may

limit the transfer of Fe between labile particulate and dissolved phases see (Hopwood et al., 2016; Lippiatt et al., 2010; Thuroczy et al., 2012)

16/13 I think these fluxes have been defined, Raiswell (et al.,) has conducted very extensive work on the different fractions of Fe present in glacially derived particles (Raiswell et al., 1994, 2010; Raiswell and Canfield, 2012) and what this means for lability. It was this early work, to my understanding, which lead to the more recent focus on the labile ferrihydrite fraction – because this is, to a first order approximation, the labile sedimentary Fe fraction which may plausibly affect primary production.

17/10... onwards. Given that models cannot agree on how important Fe-fertilization is in the present, how can you robustly conclude that the Fe source will increase in the future? I'm not sure the authors present anything that supports this statement and think the conclusion would be stronger without it. It is (unless you can produce literature to support this) presently an unsupported argument that increasing discharge will increase Fe fertilization.

Figure 1: Just to clarify, on (b) the 'day-1' means as if the flux was uniform across the year (i.e. an annual value divided by 365)? This seems a little strange way of displaying the data as presumably the actual melt rate during summer is much larger than this and for much of the year it is 0.

Figure 3: What does the white area correspond to? Maybe define, I guess something like no meaningful change?

Figure 5: I assume the colour bar should be the same as 4?

Figure 8. The caption for this figure seems to be completely incorrect.

References referred to:

Boyd, P. W., Arrigo, K. R., Strzepek, R. and Van Dijken, G. L.: Mapping phytoplankton iron utilization: Insights into Southern Ocean supply mechanisms, J. Geophys. Res. Ocean., 117, C06009, doi:10.1029/2011JC007726, 2012.

Cape, M. R., Vernet, M., Pettit, E. C., Wellner, J., Truffer, M., Akie, G., Domack, E., Leventer, A., Smith, C. R. and Huber, B. A.: Circumpolar Deep Water Impacts Glacial Meltwater Export and Coastal Biogeochemical Cycling Along the West Antarctic Peninsula , Front. Mar. Sci. , 6, 144 [online] 10.3389/fmars.2019.00144, 2019.

Duprat, L. P. A. M., Bigg, G. R. and Wilton, D. J.: Enhanced Southern Ocean marine productivity due to fertilization by giant icebergs, Nat. Geosci, 9(3), 219–221 10.1038/ngeo2633, 2016.

Hart, T. J.: Discovery Reports, Discov. Reports, VIII, 1–268, 1934.

Helly, J. J., Kaufmann, R. S., Stephenson Jr., G. R. and Vernet, M.: Cooling, dilution and mixing of ocean water by free-drifting icebergs in the Weddell Sea, Deep. Res. Part Ii-Topical Stud. Oceanogr., 58(11–12), 1346–1363, doi:10.1016/j.dsr2.2010.11.010, 2011.

Hopwood, M. J., Connelly, D. P., Arendt, K. E., Juul-Pedersen, T., Stinchcombe, M. C., Meire, L., Esposito, M. and Krishna, R.: Seasonal changes in Fe along a glaciated Greenlandic fjord, Front. Earth Sci., 4, doi:10.3389/feart.2016.00015, 2016.

Hopwood, M. J., Carroll, D., Browning, T. J., Meire, L., Mortensen, J., Krisch, S. and Achterberg, E. P.: Non-linear response of summertime marine productivity to increased meltwater discharge around Greenland, Nat. Commun., 9, 3256, doi:10.1038/s41467-018-05488-8, 2018.

De Jong, J. T. M., Stammerjohn, S. E., Ackley, S. F., Tison, J.-L., Mattielli, N. and Schoemann, V.: Sources and fluxes of dissolved iron in the Bellingshausen Sea (West Antarctica): The importance of sea ice, icebergs and the continental margin, Mar. Chem., doi:10.1016/j.marchem.2015.08.004, 2015.

Laufkötter, C., Stern, A. A., John, J. G., Stock, C. A. and Dunne, J. P.: Glacial Iron Sources Stimulate the Southern Ocean Carbon Cycle, Geophys. Res. Lett., 45, 13,377-13,385, doi:10.1029/2018GL079797, 2018.

Lippiatt, S. M., Lohan, M. C. and Bruland, K. W.: The distribution of reactive iron in northern Gulf of Alaska coastal waters, Mar. Chem., 121(1–4), 187–199, doi:10.1016/j.marchem.2010.04.007, 2010.

Maiti, K., Charette, M. A., Buesseler, K. O. and Kahru, M.: An inverse relationship between production and export efficiency in the Southern Ocean, Geophys. Res. Lett., 40(8), 1557–1561, doi:10.1002/grl.50219, 2013.

Meire, L., Meire, P., Struyf, E., Krawczyk, D. W., Arendt, K. E., Yde, J. C., Juul Pedersen, T., Hopwood, M. J., Rysgaard, S. and Meysman, F. J. R.: High export of dissolved silica from the Greenland Ice Sheet, Geophys. Res. Lett., 43(17), 9173–9182, doi:10.1002/2016GL070191, 2016.

Le Moigne, F. A. C., Henson, S. A., Cavan, E., Georges, C., Pabortsava, K., Achterberg, E. P., Ceballos-Romero, E., Zubkov, M. and Sanders, R. J.: What causes the inverse relationship between primary production and export efficiency in the Southern Ocean?, Geophys. Res. Lett., doi:10.1002/2016GL068480, 2016.

Raiswell, R. and Canfield, D. E.: The Iron biogeochemical Cycle Past and Present, Geochemical Perspect., 1(1), 1–220, doi:10.7185/geochempersp.1.1, 2012.

Raiswell, R., Canfield, D. E. and Berner, R. A.: A COMPARISON OF IRON EXTRACTION METHODS FOR THE DETERMINATION OF DEGREE OF PYRITISATION AND THE RECOGNITION OF IRON-LIMITED PYRITE FORMATION, Chem. Geol., 111(1–4), 101–110, doi:10.1016/0009-2541(94)90084-1, 1994.

Raiswell, R., Vu, H. P., Brinza, L. and Benning, L. G.: The determination of labile Fe in ferrihydrite by ascorbic acid extraction: Methodology, dissolution kinetics and loss of solubility with age and de-watering, Chem. Geol., 278(1–2), 70–79, doi:10.1016/j.chemgeo.2010.09.002, 2010.

Schwarz, J. N. and Schodlok, M. P.: Impact of drifting icebergs on surface phytoplankton biomass in the Southern Ocean: Ocean colour remote sensing and in situ

iceberg tracking, Deep. Res. Part I Oceanogr. Res. Pap., 56(10), 1727–1741, doi:10.1016/j.dsr.2009.05.003, 2009.

St-Laurent, P., Yager, P. L., Sherrell, R. M., Stammerjohn, S. E. and Dinniman, M. S.: Pathways and supply of dissolved iron in the Amundsen Sea (Antarctica), J. Geophys. Res. Ocean., doi:10.1002/2017JC013162, 2017.

St-Laurent, P., Yager, P. L., Sherrell, R. M., Oliver, H., Dinniman, M. S. and Stammerjohn, S. E.: Modeling the Seasonal Cycle of Iron and Carbon Fluxes in the Amundsen Sea Polynya, Antarctica, J. Geophys. Res. Ocean., 124(3), 1544–1565, doi:10.1029/2018JC014773, 2019.

Stephenson, G. R., Sprintall, J., Gille, S. T., Vernet, M., Helly, J. J. and Kaufmann, R. S.: Subsurface melting of a free-floating Antarctic iceberg, Deep Sea Res. Part II Top. Stud. Oceanogr., 58(11), 1336–1345, doi:https://doi.org/10.1016/j.dsr2.2010.11.009, 2011.

Thuroczy, C.-E., Alderkamp, A.-C., Laan, P., Gerringa, L. J. A., Mills, M. M., Van Dijken, G. L., De Baar, H. J. W. and Arrigo, K. R.: Key role of organic complexation of iron in sustaining phytoplankton blooms in the Pine Island and Amundsen Polynyas (Southern Ocean), Deep. Res. Part Ii-Topical Stud. Oceanogr., 71–76, 49–60, doi:10.1016/j.dsr2.2012.03.009, 2012.

Wu, S.-Y. and Hou, S.: Impact of icebergs on net primary productivity in the Southern Ocean, Cryosph., 11(2), 707–722, doi:10.5194/tc-11-707-2017, 2017.

---

## Referee Comment (RC2) · Robert Raiswell (Referee) · 25 May 2019

Person Review by Raiswell This is an excellent contribution and is entirely suitable for Biogeosciences. The authors have used a biogeochemical model to examine the delivery of Fe from the Antarctic Ice shelf. I agree with their statement that iceberg and ice shelf delivery have largely been ignored in other biogeochemical modelling studies and this is a welcome attempt to address this issue. The model produces some important new insights which will need validating in further studies, when appropriate data are available. I also agree with the authors that;  c There is considerable uncertainty in the magnitude of all the different fluxes (and this applies just as much

to atmospheric dust, as to the newer, less well-studied fluxes such as icebergs) • There are also difficulties in using the data to examine the down-stream impacts on productivity and export. The value of this paper is in recognising these issues and making sensible attempts to address them. I would hope that this study is used by the community to focus on the main areas of uncertainty, and stimulate further observational studies. Especially as a particular difficulty the authors faced is that there are few relevant iceberg data sets and, in fact, there are more observational data from the Greenland ice-hosted sources than from the AIS. I commend the authors on reviewing the literature so thoroughly. I emphasize that I lack the expertise to comment on the models in detail but other comments below are keyed to page and line numbers. Page 2, line 3. The Raiswell reference is not the best as there are numerous studies of dust deposition to the SO. However the Raiswell data (I hope!) is more useful than many others because the extraction used relates to mineralogy, and specifically to ferrihydrite which is potentially the most bioavailable mineral form. You could move the Tagliabue ref to after 'SO' and before the colon, and then maybe cite a Boyd reference, perhaps the Mar Chem 2010 paper. The Raiswell reference would be better in the iceberg citations. Page 2, line 25. Delete 'through finely ground rocks'. The rock source is larger than the dissolved sources but the latter is not negligible and may be the most bioavailable. Page 2, line 27. Add 'fueling productivity in surface waters'. Page 3, line 3. Unfortunately the 50 samples are largely from Greenlandic icebergs and not Antarctica. Clarify this. Page 3, line 14. The impacts on productivity are the point at which my biological expertise starts to fail. The impact critically depends on how Fe affects on productivity and thus carbon export. The authors obviously need to explore this issue but an expression of caution would be wise. Maybe add 'cycles, depending on how Fe inputs relate to productivity and carbon export'. Page 3, line 18. I welcome the attempt to consider vertical distributions of iceberg Fe and their influence on the surrounding seawater. No doubt the distributions will turn out to be very variable, not least because the vertical iceberg Fe contents will alter as icebergs overturns. Page 3, line 25. 10% is OK but probably conservative. I would think that most ferrihydrite

would be bioavailable, especially as ferrihydrite carries a significant fraction of ferrous iron. There is a brief discussion of this in my recent Frontiers paper, v. 6. No 222, doi: 10.3389/feart.2018.00222. You might be interested to look at this and at the EPSL 493, 92-101 paper by Hawkings. The Frontiers paper also raises the issue that ice is not inert and is able to catalyse the reduction of ferrihydrite. Also the freezing of sea ice produces pockets of Fe-enriched, chloride complexed brines that would be released early in melting. I am not suggesting that you need to cite these papers, I am only making the point, as you realise, that there are many areas of uncertainty which could profoundly alter the bioavailability percentage. It might be worth stating that you have not considered ice-water-mineral reactions. Page 4, line 23. Add wt.% after data. Page 4, line 30. Reword as 'no observational data are available that allow the shelf Fe fluxes from Antarctica to be constrained, as..' There is a very crude estimate of 5.3 Gmoles/yr in Raiswell et al (2016) Page 5, line 5. It would be good to have a table showing the fluxes and solubilities assumed for dust, sediments and sea ice in the CTL model. Page 6, line 30. This states that the 1.5 and 6.3 nmol/L values are over and above the CTL data. Can the authors clarify what is being derived here? I think the models produce 'dissolved Fe' (see the discussion in the Raiswell Frontiers paper). In any event the data would have to be compared with seawater measurements on water filtered through 0.45 micron, which is 'dissolved Fe'. These model values would be at the upper limit of actual seawater 'dissolved Fe' concentrations outside of coastal regions. Page 7, line 9. Sentence unclear. Page 7 line 30. The caption to fig. 5 needs to clarify which are the positive and negative areas. Page 8 line 17. The potential of this deep reservoir is one of the important insights that your study produces. Page 10, line 10 on. This seems reasonable. The whole point about icebergs is that they can transport, which is not true for ice shelf sources. But it is good to see this confirmed. Page 10, line 24. My figure 8 shows the difference in surface Fe concentrations, not chlorophyll. Has a diagram been incorrectly inserted? Page 11, line 11. Delete 'the' before Bouvet island. Paage 13, line 30 on. I agree that this difference is hard to understand but you make a crucial point; that modelling the ice-hosted sources is at present
difficult; although the attempt is certainly valuable (see above). Page 15, line 25 on. Yes, delivery will vary as iceberg melting occurs. Page 16, line 5. I would prefer to be cautious here and describe the most labile source as' potentially bioavailable'. But I agree that there will be a range of Fe mineral reactivities each with different rates of reaction or dissolution or grazing interactions, and thus different bioavailabilities. Page 17, line 2. This is another useful finding, although again not unexpected that iceberg effects are spatially variable.

———————————————————

---

## Referee Comment (RC3) · Anonymous Referee #3 · 20 Jun 2019

This paper presents a study evaluating the impact of Fe supply from Antarctic ice shelves and icebergs on productivity/chlorophyll in the Southern Ocean. It presents a thorough examination of the uncertainties associated with the fertilisation capacity of this input and highlights remaining differences between the observations and model results even when these Fe sources are included. The authors highlight particular areas where existing models can be improved, or futher in-situ observations are required. With some improvements, I believe this paper is a valuable addition to the field.

I am reviewing this paper with shallow knowledge of the biogeochemistry and will be focusing on iceberg and ice shelf melt.

Larger corrections

It was not clear whether the Fe supply is injected at a particular layer, and no further dynamics apply, or whether once the Fe is added, those waters are able to mix (as is likely to happen associated with the buoyancy injection from meltwater)? This applies throughout the paper, but in particular on page 13 (line 30-35) where you discuss the possible cause of differences between your primary production are that found in Laufkötter et al (2018).

Some further discussion of this, and the general background associated with the melt-water pump would be valuable. Recent papers have shown the effect of this in Antarctic waters (St-Laurent et al., 2017, 2019: Cape et al., 2019) and in your discussion you only refer to this process associated with Greenland glaciers (pg 15, line 21).

Similar to the meltwater pump model for ice shelves, are similar processes considered for iceberg melt? For iceberg melt occurring at depth, mixing with surrounding waters may result in upwelling of nutrient-rich waters, rather than the iceberg Fe-source remaining trapped below the ML.

Smaller corrections

Abstract: Line 12-13: The comment that seasonal variations have regional impacts that are then "almost negligible" is slightly confusing. May be better to re-word this sentence?

Pg2: Some other references to consider in this section are Cape et al (2019) (ice shelf meltwater pump), Biddle et al (2015), in-situ observations of productivity from iceberg melt,

Line 17: I'm not sure you've defined AIS yet. Be very clear about the differences between AIS (I assume Antarctic Ice Sheet?), ice shelves and icebergs.

Line 27: "fueling" in what way? Is the Fe used, or is it just supplied? Line 34: remove "the" before "Prydz Bay"

Pg 3, line 18: I would read "along the water column" as along the iceberg tracks (spatial/horizontal). Is this what you mean, or do you mean the vertical distribution?

Pg 4, line 10: For those unfamiliar with the model, a brief description here of how the freshwater fluxes are added would be helpful. Are the ice cavities simulated? Or is it a vertical wall in the model that freshwater/Fe is added through? In the latter case, what does "between the base and the grounding line of the ice shelves" then refer to – freshwater fluxes are equally added between the depth of the ice shelf (say 400 m) and the seabed? In this situation, many recent papers have shown that the strongest outflow is at the base of the ice shelf and diminishes with depth, in addition to buoyant upwelling to the surface (Naveira Garabato et al., 2017; Nakayama et al., 2014). Again, this is relevant to the meltwater pump.

Pg 6, line 24: "as well as in the Ross Sea until the Amundsen Sea" – I'm not sure what you mean by this? The Indian and Pacific sectors include these coasts? (See comment in figures about specifying what region you are referring to).

Pg7, line 9-10: I am not sure what you mean by "Furthermore, in winter…".

Pg 10, Lines 14-18: I think the meltwater pump should be included here – the ice shelf Fe is not just injected deeper than the mixed layer. Line 33: "The mains" → "The main"

Pg 11, Line 11: remove "the" in front of Bouvet. Line 15: remove "by" in front of "∼1.3…"

Pg 13, Line 30-35: This deserves more discussion about why there are differences between the models with similar Fe fluxes. Are there physical differences in the models in how they treat mixing of meltwater/depth of meltwater input?

Pg 15, Line 20-23: This seems quite likely (e.g. Cape et al, 2019) – see earlier general comment. Line 34: "we did not explore"

Figures – I would like the labels on the maps for longitudes to be slightly larger, and to be consistent with the direction/order of labelling panels. You also refer to the different

sectors a lot (e.g. Indian-Pacific sector) – is it possible to mark the boundaries of these sectors, perhaps just on the first figure?

Figure 5 – what is the colorbar for this figure?

Figure 8 has an incorrect caption (it is identical to Figure 7).

References:

Biddle, L. C., J. Kaiser, K. J. Heywood, A. F. Thompson, and A. Jenkins (2015). Ocean glider observations of iceberg-enhanced biological production in the northwestern Weddell Sea. Geophys. Res. Lett., 42, doi:10.1002/2014GL062850.

Cape, M. R., Vernet, M., Pettit, E. C., Wellner, J., Truffer, M., Akie, G., Domack, E., Leventer, A., Smith, C. R. and Huber, B. A (2019). Circumpolar Deep Water Impacts Glacial Meltwater Export and Coastal Biogeochemical Cycling Along the West Antarctic Peninsula , Front. Mar. Sci. , 6, 144 [online] 10.3389/fmars.2019.00144.

Naveira Garabato, A. C., A. Forryan, P. Dutrieux, L. Brannigan, L. C. Biddle, K. J. Heywood, A. Jenkins, Y. L. Firing and S. Kimura (2017). Vigorous lateral export of the meltwater outflow from beneath an Antarctic ice shelf. Nature 542, 219–222, doi:10.1038/nature20825

St-Laurent, P., Yager, P. L., Sherrell, R. M., Stammerjohn, S. E. and Dinniman, M. S.: Pathways and supply of dissolved iron in the Amundsen Sea (Antarctica) (2017). J. Geophys. Res. Ocean., doi:10.1002/2017JC013162.

St-Laurent, P., Yager, P. L., Sherrell, R. M., Oliver, H., Dinniman, M. S. and Stammerjohn, S. E (2019). Modeling the Seasonal Cycle of Iron and Carbon Fluxes in the Amundsen Sea Polynya, Antarctica, J. Geophys. Res. Ocean., 124(3), 1544–1565, doi:10.1029/2018JC014773.

---

## Author Comment (AC1) · 17 Jul 2019

**Response to Reviewer Robert Raiswell**

Person Review by Raiswell. This is an excellent contribution and is entirely suitable for Biogeosciences. The authors have used a biogeochemical model to examine the delivery of Fe from the Antarctic Ice shelf. I agree with their statement that iceberg and ice shelf delivery have largely been ignored in other biogeochemical modelling studies and this is a welcome attempt to address this issue. The model produces some important new insights which will need validating in further studies, when appropriate data are available. I also agree with the authors that; 1) There is considerable uncertainty in the magnitude of all the different fluxes (and this applies just as much to atmospheric dust, as to the newer, less well-studied fluxes such as icebergs), 2) There are also difficulties in using the data to examine the down-stream impacts on productivity and export. The value of this paper is in recognising these issues and making sensible attempts to address them. I would hope that this study is used by the community to focus on the main areas of uncertainty, and stimulate further observational studies. Especially as a particular difficulty the authors faced is that there are few relevant iceberg data sets and, in fact, there are more observational data from the Greenland ice-hosted sources than from the AIS. I commend the authors on reviewing the literature so thoroughly.

I emphasize that I lack the expertise to comment on the models in detail but other comments below are keyed to page and line numbers.

**We thank Robert Raiswell for his review and general support for our manuscript. We present our response in bold and preceded by '>' in case of formatting errors.**

Page 2, line 3. The Raiswell reference is not the best as there are numerous studies of dust deposition to the SO. However the Raiswell data (I hope!) is more useful than many others because the extraction used relates to mineralogy, and specifically to ferrihydrite which is potentially the most bioavailable mineral form. You could move the Tagliabue ref to after 'SO' and before the colon, and then maybe cite a Boyd reference, perhaps the Mar Chem 2010 paper. The Raiswell reference would be better in the iceberg citations.

**> Acknowledged and addressed**

Page 2, line 25. Delete 'through finely ground rocks'. The rock source is larger than the dissolved sources but the latter is not negligible and may be the most bioavailable.

**> Acknowledged and addressed**

Page 2, line 27. Add 'fueling productivity in surface waters'.

**We mean "delivering Fe".**

**We modified the sentence to clarify this point as follows:**

**"The melting of icebergs and ice shelves releases Fe to seawater as particulate, dissolved, and potentially dissolvable forms fueling the water column in Fe"**

Page 3, line 3. Unfortunately the 50 samples are largely from Greenlandic icebergs and not Antarctica. Clarify this.

**Clarified in the article**

Page 3, line 14. The impacts on productivity are the point at which my biological expertise starts to fail. The impact critically depends on how Fe affects on productivity and thus carbon export. The authors obviously need to explore this issue but an expression of caution would be wise. Maybe add 'cycles, depending on how Fe inputs relate to productivity and carbon export'.

**Expression of caution added in the article**

Page 3, line 18. I welcome the attempt to consider vertical distributions of iceberg Fe and their influence on the surrounding seawater. No doubt the distributions will turn out to be very variable, not least because the vertical iceberg Fe contents will alter as icebergs overturns.

Page 4, line 25. 10% is OK but probably conservative. I would think that most ferrihydrite would be bioavailable, especially as ferrihydrite carries a significant fraction of ferrous iron. There is a brief discussion of this in my recent Frontiers paper, v. 6. No 222, doi: 10.3389/feart.2018.00222. You might be interested to look at this and at the EPSL 493, 92-101 paper by Hawkings. The Frontiers paper also raises the issue that ice is not inert and is able to catalyse the reduction of ferrihydrite. Also the freezing of sea ice produces pockets of Fe-enriched, chloride complexed brines that would be released early in melting. I am not suggesting that you need to cite these papers, I am only making the point, as you realise, that there are many areas of uncertainty which could profoundly alter the bioavailability percentage. It might be worth stating that you have not considered ice-water-mineral reactions.

**We thank the reviewer for drawing our attention to these two papers.**
**Text modified**

Page 4, line 23. Add wt.% after data.

**> Acknowledged and addressed**

Page 4, line 30. Reword as 'no observational data are available that allow the shelf Fe fluxes from Antarctica to be constrained, as..' There is a very crude estimate of 5.3 Gmoles/yr in Raiswell et al (2016)

**Here we mean "Antarctic ice shelf Fe fluxes" and not "Antarctic shelf Fe fluxes".**
**Sentence reworded.**

Page 5, line 5. It would be good to have a table showing the fluxes and solubilities assumed for dust, sediments and sea ice in the CTL model.

**Fe fluxes from other sources simulated in the CTL experiment added in table 1.**

Page 6, line 30. This states that the 1.5 and 6.3 nmol/L values are over and above the CTL data. Can the authors clarify what is being derived here? I think the models produce 'dissolved Fe' (see the discussion in the Raiswell Frontiers paper). In any event the data would have to be compared with seawater measurements on water filtered through 0.45 micron, which is 'dissolved Fe'. These model values would be at the upper limit of actual seawater 'dissolved Fe' concentrations outside of coastal regions.

**The model values are concentrations in dissolved Fe.**
**Due to the poor availability of data in the Atlantic plume northeast of the Antarctic Peninsula, it is difficult to compare to real concentrations. However, it is true that these concentrations are probably at the upper limit of Fe concentrations in the open ocean but still potentially realistic in coastal regions (de Jong et al., 2012).**

Page 7, line 9. Sentence unclear.

**Sentence modified as follows:**
**"Furthermore, in winter, deep mixing entrained to the surface Fe that was released in summer below the euphotic zone and that escaped consumption by phytoplankton due to the lack of light."**

Page 7 line 30. The caption to fig. 5 needs to clarify which are the positive and negative areas.

**Caption modified in order to clarify this point.**

Page 8 line 17. The potential of this deep reservoir is one of the important insights that your study produces.

Page 10, line 10 on. This seems reasonable. The whole point about icebergs is that they can transport, which is not true for ice shelf sources. But it is good to see this confirmed.

Page 10, line 24. My figure 8 shows the difference in surface Fe concentrations, not chlorophyll. Has a diagram been incorrectly inserted?

**The right caption for Figure 8 is:**
**"Surface chlorophyll concentrations in summer (December, January, and February) from (a) satellite observations (MODIS-Aqua, Johnson et al., (2013)), (b) the CTL experiment, and (c) the SOLUB5 experiment in the Southern Ocean, south of 50° S.**

Page 11, line 11. Delete 'the' before Bouvet island.

**> Acknowledged and addressed**

Page 13, line 30 on. I agree that this difference is hard to understand but you make a crucial point; that modelling the ice-hosted sources is at present difficult; although the attempt is certainly valuable (see above).

Page 15, line 25 on. Yes, delivery will vary as iceberg melting occurs.

Page 16, line 5. I would prefer to be cautious here and describe the most labile source as' potentially bioavailable'. But I agree that there will be a range of Fe mineral reactivities each with different rates of reaction or dissolution or grazing interactions, and thus different bioavailabilities.

**OK replaced in the text.**

Page 17, line 2. This is another useful finding, although again not unexpected that iceberg effects are spatially variable.

**References**

de Jong, J., Schoemann, V., Lannuzel, D., Croot, P., de Baar, H., & Tison, J.-L. (2012). Natural iron fertilization of the Atlantic sector of the Southern Ocean by continental shelf sources of the Antarctic Peninsula. *Journal of Geophysical Research: Biogeosciences*, *117*(G1), n/a-n/a. https://doi.org/10.1029/2011JG001679

Johnson, R., Strutton, P. G., Wright, S. W., McMinn, A., & Meiners, K. M. (2013). Three improved satellite chlorophyll algorithms for the Southern Ocean. *Journal of Geophysical Research: Oceans*, *118*(7), 3694–3703. https://doi.org/10.1002/jgrc.20270

---

## Author Comment (AC2) · 17 Jul 2019

**Response to reviewer #1**

The authors present a model study investigating how Fe input into the Southern Ocean from icebergs and the Antarctic Ice Sheet affects the distribution of Fe and primary production in the marine environment. Recognizing the uncertainty in the magnitude and nature of these Fe sources, and thus several difficulties in meaningfully parametrizing them to date, the authors opt to model several scenarios with important differences in Fe solubility and the distribution of melt-derived Fe in the water column. The results, with respect to primary production and C export, fall within the (very broad) range of other model studies suggesting a modest impact of this Fe on Southern Ocean productivity. A key strength of this specific study is that it makes considerable effort to highlight the many uncertainties surrounding this Fe source. Numerous other recent works have proposed much stronger effects but neglected to consider some, or all, of the uncertainties highlighted herein. Whilst there are a few areas in the text where I think some improvements can be made, I generally therefore consider this to be a valuable addition to the field, suitable for publication in BGS and, in my opinion, one of the most comprehensive manuscripts on the subject of modelling these Fe fluxes to date.

My expertise is in biogeochemistry, I defer to a more qualified reviewer for issues concerning details of the model used. Before returning the text to the journal, it would benefit slightly from a read through from an English editor.

**We thank reviewer #1 for his detailed review and general support for our manuscript.**
**We present our response in bold and preceded by '>' in case of formatting errors.**

General comment; have the authors considered the meltwater 'pump' effect outlined in some recent work (see comment on page 4, (Cape et al., 2019; St-Laurent et al., 2017, 2019)? I wasn't clear if this effect would be captured in the model or not.

**In our model configuration, the cavities below the ice shelves are not opened. To mimic the overturning circulation driven by these unresolved ice shelves, we used the parametrisation of Mathiot et al. (2017) which prescribes a meltwater flux of ice shelf uniformly distributed over the depth and width of the unresolved cavity opening, from the mean ice front draft down to the seabed, or the grounding line depth if it is shallower. Mathiot et al. (2017) showed that this parametrisation of the ice shelf melting drives a buoyant overturning circulation along the coast, i.e. the meltwater pump, similar to that simulated by cavities when they are explicitly resolved.**

General comment: How is C export scaled to primary production in the model, does the model successfully replicate the observed relationship between the two? Looking at some other models and calculations in the literature, it appears to me that a key reason why very broad ranges are often quoted for C export from

specific Fe fertilization scenarios is simply because of the way Fe or productivity/chlorophyll a is scaled to C export. The 'high' C export estimate of (Duprat et al., 2016) is scaled linearly with chlorophyll/productivity –which is not consistent with observational Southern Ocean data. It is not clear to me if this is also a problem with the (Laufkötter et al., 2018) model which matches the Duprat calculation surprisingly well producing a fertilizing effect significantly above that found herein. (Observations with multiple methods show that C export efficiency declines sharply with increasing productivity in the Southern Ocean, although the precise reason(s) for this seem to be unclear (Maiti et al., 2013; Le Moigne et al., 2016).

**We completely agree with this comment. But in the actual context of non-consensus about the export ratio in the Southern Ocean, it is very difficult to estimate whether our model replicate "realistically" the observed relationship between C export and primary productivity due to the poor data spatial and temporal coverage. In our model, the relationship between PP and C export does not show a linear pattern as illustrated in Figure 1. Nevertheless, there is a clear trend that shows higher export with higher primary productivity which is highly variable at the local and temporal scale. We don't know if in the COBALT model used by (Laufkötter et al., 2018), the relationship is different which could explain the differences. In fact, a detailed and thorough comparison with that study is really challenging because we lack many information that would be necessary. These differences are really intriguing and would probably deserve a careful analysis involving a collaboration between the two groups.**

[Figure]

Fig 1. Density relationship between primary production and C export at 150 m depth over the Southern Ocean, south of 50°S, in the SOLUB5 experiment.

Specific comments by Page/line Title: Antarctic Ice Sheet

**> Acknowledged and addressed**

1/12 'seasonal variations' in the timing of melting? If I understand correctly, this sentence would read better 'Seasonal variations make almost negligible differences...'

**To clarify a possible misunderstanding, we modified the sentence as follows :**
**"seasonal variations of the iceberg Fe fluxes have regional impacts which are small for annual-mean primary productivity and C export at the scale of the SO"**

2/3 Raiswell 2016 does not contain extensive atmospheric dust work, I am sure there are better values/references for dust deposition

**Other references cited for dust deposition.**

2/14 'the mean flux'. You mean the total flux?

**We mean the total mean flux. Modified.**

2/16 'the few modelling studies conducted to date''

**> Acknowledged and addressed**

2/27 'fueling surface waters'. You mean 'fueling' productivity or just delivering Fe?

**We mean "delivering Fe". We modified the sentence to clarify this point as follows:**
**"The melting of icebergs and ice shelves releases Fe to seawater as particulate, dissolved, and potentially dissolvable forms fueling the water column in Fe"**

2/28 Not sure this is accurate, it has been speculated that glacially derived Fe was fueling primary production in the Southern Ocean for some time e.g. (Hart, 1934), it just has proved very difficult to quantify.

**We thank reviewer #1 to introduce this reference.**
**Text modified in accordance.**

2/32 See also (Wu and Hou, 2017) - a particularly interesting read as it, when compared to (Duprat et al., 2016) demonstrates that there are significant differences in observational data constraining the effect of icebergs, not just in the models.

**OK.**

2/34 'the Prydz bay'. Delete the

**> Acknowledged and addressed**

3/12 : : :will increase the supply of Fe: : : Assuming that the Fe input scales linearly with ice-melt, which may be a little speculative

**We agree with this comment.**
**Sentence changed as follows:**
**"The projected decline of the AIS will potentially increase the release of Fe from icebergs and ice shelves in the SO with possible significant impacts on marine productivity and biogeochemical cycles, depending on how Fe inputs relate to productivity and carbon export."**

3/18 'along the water column' means horizontal, you mean 'through'

**> Acknowledged and addressed**

4/10 Here something concerning the 'meltwater pump' may be relevant. High Fe concentrations adjacent to Ice sheets (in the ocean) would generally be attributed to direct input from melt/sediment release etc, but release of meltwater can also 'pump' ambient to the surface and thus bring Fe from shelf sediments and the sub-surface Fe reservoir into surface waters. These effects are difficult to tease-apart from field data. But some model calculations suggest that the magnitudes of Fe from 'pumping' and from direct input (melt/freshwater/freshwater derived particles) are comparable – all be it with large uncertainties. See (Cape et al., 2019; St-Laurent et al., 2017, 2019) for overviews of this effect and what we do/don't know about it.

**Please, see our answer to general concerns.**
**Text modified to detail that the parametrisation of the ice shelf melting from Mathiot el at. (2017) simulates the buoyant overturning circulation along the coast and the associated meltwater pump.**

4/23 It is not clear to me what the % here refer to, I guess the % weight of sediment which is ferrihydrite, but please clarify (also specifically what is the mean – a global mean??)

**wt.% added after data**
**Here we mean the mean content of the estimated range from Raiswell et al. (2016)**
**Modified in the text.**

4/24 This is a muddled concept in the field in general. All labile Fe could be potentially 'biologically

available' if processed/delivered in the right way, I would stick specifically with the 'soluble' fraction rather than trying to define a 'biologically available' fraction as this is an arbitrary exercise. The concept of 'utilization' (Boyd et al., 2012) is perhaps more useful as 'bioavailability' is a qualitative term.

**We agree with Reviewer #1 that the concept of bioavailability is rather vague. Bioavailability depends on numerous factors such as the nature of the iron particles, the interactions with the ligands, the environmental conditions, … As a consequence, the fraction of iron that can be ultimately available to phytoplankton (and bacteria) is highly variable and very difficult to infer. Boyd et al. (2012) have studied the Fe utilization by phytoplankton based on observed Fe/Chl ratios. They compared this utilization to the magnitude of different sources (dust, sediment resuspension/mobilization, meltwater, …) to evaluate if these sources are related to a higher Fe utilization. The concept of utilization is thus very useful to qualitatively investigate the potential fertilization effect of different iron sources. However, this remains qualitative and based on many assumptions, among which the values of the Fe/Chl ratios are among the highest. Furthermore, the comparison to supply mechanisms still requires the definition of a bioavailable iron fraction to evaluate the magnitude of the sources. Finally, in a prognostic model, utilization is prognostically predicted based in part on the amount of iron that is available which turns back to the definition of bioavailable iron.**

4/30 Seems like an odd thing to say. 'no data allow the constraining of: : :' or 'allow us to'

**Sentence reworded as follows:**
**"no observational data are available that allow the ice shelf Fe fluxes to be constrained, as the Antarctic estimates from..."**

5/30 The 'buoyancy effect' is widely attributed with bringing iceberg-derived components (e.g. particles/Fe) to the surface, but as far as I'm aware there isn't much clear evidence of it actually doing this, or even much data to show how ice melt behaves in the real world. An alternative argument is that something akin to convective cells develop up the sides of the iceberg, and that these reach neutral buoyancy before they reach the surface i.e. most melt doesn't 'rise' to the surface. In any case, there is certainly very limited data to show how ice melt behaves around icebergs (Helly et al., 2011; Stephenson et al., 2011).

**OK. We thank the reviewer for drawing our attention to these papers.**

6/30 How do these concentrations compare to 'real' Fe concentrations in these areas?

**Due to the poor availability of data in the Atlantic plume northeast of the Antarctic Peninsula, it is difficult to compare to real concentrations. However, these concentrations are probably at the upper limit of Fe concentrations in the open ocean but potentially realistic in coastal regions (de Jong et al.,**

**2012)**.

7/9&10 This line 'Furthermore, in winter,: : :' does not make sense

**Sentence modified as follows:**
**"Furthermore, in winter, deep mixing entrained to the surface Fe that was released in summer below the euphotic zone and that escaped consumption by phytoplankton due to the lack of light."**

7/6 These concentrations are not feasible, how is scavenging constrained? Such a high dissolved Fe concentration (27 nM) would, practically immediately, precipitate.

**In coastal regions, Fe concentrations can be very high as shown in the article of de Jong et al. (2012) with measured surface Fe concentrations up to 50 nmol L$^{-1}$.**

3.1.4 Whilst the effect is poorly defined, the meltwater 'pump' should be at least mentioned here.(St-Laurent et al., 2017, 2019)

**Text modified to mention the meltwater pump.**

10/33 The mains

**> Acknowledged and addressed**

11/11 the Bouvet Island. Delete 'the'

**> Acknowledged and addressed**

11/20 Nevertheless, though small

**> Acknowledged and addressed**

11/25 CHL at the blooming season, you mean 'throughout' or 'during the season' (general comment CHL is, at a glance, similar to CTL, maybe use 'Chl a' or similar)

**We mean during the season. Modified.**
**We choose SChl for surface chlorophyll concentrations instead of CHL**

12/30 equal to

**> Acknowledged and addressed**

12/33 'are almost unchanged.' Compared to?

**Compared to the CTL experiment.**
**Added in the sentence.**

13/27 'leads to a significant increase in'

**> Acknowledged and addressed**

13/32 Indeed. The first thing I did after reading this study was to refer to (Laufkötter et al., 2018). I was very surprised to find that both studies use very similar parameterizations for the total Fe input. As a biogeochemist, my simplistic conclusion is therefore that these results (collectively) are not reproducible between models, as completely opposite conclusions are reached using practically the same Fe input. More surprising is that the results of the studies don't even overlap- given that both studies use very broad ranges in Fe input which were designed to span all environmentally relevant scenarios. This is problematic, because it makes the studies (again, collectively-this is not a specific critique of this study) impossible to interpret from a biogeochemical perspective. So the critical question is why is there such a large difference? The authors herein do a generally good job of discussing the differences between existing iceberg models, but perhaps this information (presently in the text) could be thinned a little and compiled in the form of a table which would at least eliminate some causes of differences between independent models. As a biogeochemist it is difficult to comment further other than to raise a flag that model results should be treated with extreme caution until some consensus can be found between different model studies.

**Please see also our answer to the general concerns.**
**We totally agree with your last comment and we will modified the conclusion section in accordance to this point.**

14/2 Yes, but be careful here concerning 'regionally sig. C export'. Compare (Wu and Hou, 2017) and (Schwarz and Schodlok, 2009) with (Duprat et al., 2016), the later study claims a much larger effect, but only in the C export calculated, I suspect this is largely because of how the observed data (chlorophyll) is scaled to C export and thus reflects different assumptions in the calculation rather than actual differences in the raw data.

14/29 Does (De Jong et al., 2015) not conclude that much of the Fe is sub-surface?

**Yes, this is their main conclusion regarding the iceberg Fe delivery.**

15/19 'runoff' [as a macronutrient source] this is a bit of a misleading statement, even in the North Atlantic, where macronutrient concentrations are much lower in the mixed layer, runoff dilutes the concentration of N and P macronutrients (Meire et al., 2016), so a missing macronutrient-runoff source couldn't plausibly explain the problem herein. Similarly ice contains very low macronutrient concentrations.

**Supply mechanism removed.**

15/22 This seems more plaussible, see for example (Cape et al., 2019), although even these 'upwelled' nutrient fluxes would be modest and I doubt sufficient to explain the model problem-plus they would come with Fe. In these references here, I think the authors mean (Hopwood et al., 2018) rather than the Hopwood paper listed. Alternatively, how scavenging is accounted for in the model (a difficult thing to do) presumably could cause this effect, if Fe is removed a little too slowly, it will 'over-fertilize' in the model world and thus, all other things being equal, drawdown macronutrients much faster than would be the case otherwise. As noted, I am not a model expert, but I would guess that macronutrient distributions in the model match real data better than Fe distributions and thus would speculate that problems are more likely to arise from how Fe is parametrized than with macronutrient sources/sinks.

**Reviewer #1 is correct in the fact that macronutrient distributions are better simulated by models, including ours, than Fe distribution. This is illustrated in the reference paper of PISCES (Aumont, et al., 2015). Models tend to have difficulties at properly simulating the iron distribution in the ocean ((Tagliabue et al., 2016) even if PISCES tends to perform quite well in comparison with other models that participated to the FeMIP exercise. The drawdown of nutrients close to the coast is explained by an intense primary productivity that drives an intense export of carbon and nutrients. Due to the lack of data, primary productivity is difficult to evaluate as well as chlorophyll values. We have to rely on satellite-retrieved values which may be biased in that specific region and in areas closed to the coast. This comparison indicates that we don't hugely overestimates chlorophyll levels even if they tend to be too high on average. Thus, a too intense fertilization by iron may be part of the explanation, either because scavenging is too low and/or iron input from sediments and ice shelves is too large. Another probable reason is that export is too large and efficient in the model in that region. However, due to the lack of data, this proves to be impossible to investigate properly.**

**You are right, we mean (Hopwood et al., 2018).**
**Reference modified.**

16/2 See also (Boyd et al., 2012) – specifically the 'utilization' of Fe shifts significantly along 'Iceberg Alley'

**Effectively, their results suggest that the rates of iron utilization appear to be considerably less than that potentially supplied from iceberg melting along their drift. They also revealed the impossibility to evaluate to which extent because of the contributions from other sources of Fe (sediment and dust) in this region. Moreover, the Fe utilization was computed from the net primary production derived from satellite products which might be potentially severely biased. Indeed, in the Southern Ocean, satellite products were pointed out to particularly underestimate chl a concentrations (Johnson et al., 2013), and inferred net primary production are associated with very large uncertainties (Saba et al., 2011).**

**The shift in the "utilization" of Fe from iceberg of Boyd et al. (2012) has been added in this section.**

16/7 Perhaps, but then this becomes a question of organic ligands and to what extent these are able to transfer Fe into the dissolved phase. I'm not aware of any work around icebergs looking at ligand-iceberg interactions, but this has been investigated with respect to glacially derived particles, for general discussion of how ligands may limit the transfer of Fe between labile particulate and dissolved phases see (Hopwood et al., 2016; Lippiatt et al., 2010; Thuroczy et al., 2012)

**Ligands clearly control the amount of iron that can remain in the soluble fraction when particles released by icebergs and ice shelves dissolve in sea water. The studies mentioned by Reviewer #1 show that meltwater contains quite significant amounts of ligands that increase the amount of iron that can dissolve or remain dissolved. As a consequence, the apparent solubilization of glacial particles is controlled partly by these ligands. In our model, we don't include a potential source of ligands from meltwater because as said by Reviewer #1, we do not have any data to constrain that input. Thus, the ligands concentration in the vicinity of the icebergs is supposed to be identical to that of the open ocean. If meltwater is an important source of ligands, this would mean that our model is underestimating the supply of soluble iron from icebergs (and ice shelves).**

16/13 I think these fluxes have been defined, Raiswell (et al.,) has conducted very extensive work on the different fractions of Fe present in glacially derived particles (Raiswell et al., 1994, 2010; Raiswell and Canfield, 2012) and what this means for lability. It was this early work, to my understanding, which lead to the more recent focus on the labile ferrihydrite fraction – because this is, to a first order approximation, the labile sedimentary Fe fraction which may plausibly affect primary production.

**OK. We thank the reviewer for drawing our attention to these papers.**

17/10: : : onwards. Given that models cannot agree on how important Fe-fertilization is in the present, how can you robustly conclude that the Fe source will increase in the future? I'm not sure the authors present anything that supports this statement and think the conclusion would be stronger without it. It is (unless you

can produce literature to support this) presently an unsupported argument that increasing discharge will increase Fe fertilization.

**You are right, however a mechanistic increase in the AIS supply will at least increase surface Fe concentrations in the model. We might better say that as no agreement are found between models in their biogeochemical response to the AIS Fe supply, it is for now impossible to evaluate the impacts of climate change on this external source of Fe and their consequences on marine biogeochemistry in the Southern Ocean. This also points to the necessity to understand the mechanisms that explain the very large differences that are simulated by the models.**

Figure 1: Just to clarify, on (b) the 'day-1' means as if the flux was uniform across the year (i.e. an annual value divided by 365)? This seems a little strange way of displaying the data as presumably the actual melt rate during summer is much larger than this and for much of the year it is 0.

**We modified Figure 1 to express the AIS Fe fluxes in kg m$^{-2}$ yr$^{-1}$.**

Figure 3: What does the white area correspond to? Maybe define, I guess something like no meaningful change?

**Caption modified as suggested.**

Figure 5: I assume the colour bar should be the same as 4?

**It is the same colour bar as in Figure 4. Added to Figure 5.**

Figure 8. The caption for this figure seems to be completely incorrect.

**Here, the right caption is: "Surface chlorophyll concentrations in summer (December, January, and February) from (a) satellite observations (MODIS-Aqua, Johnson et al., (2013)), (b) the CTL experiment, and (c) the SOLUB5 experiment in the Southern Ocean, south of 50° S.**

**References**

Aumont, O., Ethé, C., Tagliabue, A., Bopp, L., & Gehlen, M. (2015). PISCES-v2: An ocean biogeochemical model for carbon and ecosystem studies. *Geoscientific Model Development*, *8*(8), 2465–2513. https://doi.org/10.5194/gmd-8-2465-2015

de Jong, J., Schoemann, V., Lannuzel, D., Croot, P., de Baar, H., & Tison, J.-L. (2012). Natural iron fertilization of the Atlantic sector of the Southern Ocean by continental shelf sources of the Antarctic

Peninsula. *Journal of Geophysical Research: Biogeosciences*, *117*(G1), n/a-n/a. https://doi.org/10.1029/2011JG001679

Hopwood, M. J., Carroll, D., Browning, T. J., Meire, L., Mortensen, J., Krisch, S., & Achterberg, E. P. (2018). Non-linear response of summertime marine productivity to increased meltwater discharge around Greenland. *Nature Communications*, *9*(1). https://doi.org/10.1038/s41467-018-05488-8

Johnson, R., Strutton, P. G., Wright, S. W., McMinn, A., & Meiners, K. M. (2013). Three improved satellite chlorophyll algorithms for the Southern Ocean. *Journal of Geophysical Research: Oceans*, *118*(7), 3694–3703. https://doi.org/10.1002/jgrc.20270

Laufkötter, C., Stern, A. A., John, J. G., Stock, C. A., & Dunne, J. P. (2018). Glacial Iron Sources Stimulate the Southern Ocean Carbon Cycle. *Geophysical Research Letters*, *45*(24), 13,377-13,385. https://doi.org/10.1029/2018GL079797

Mathiot, P., Jenkins, A., Harris, C., & Madec, G. (2017). Explicit representation and parametrised impacts of under ice shelf seas in the z$\pi$ coordinate ocean model NEMO 3.6. *Geoscientific Model Development*, *10*(7), 2849–2874. https://doi.org/10.5194/gmd-10-2849-2017

Raiswell, R., Hawkings, J. R., Benning, L. G., Baker, A. R., Death, R., Albani, S., … Tranter, M. (2016). Potentially bioavailable iron delivery by iceberg-hosted sediments and atmospheric dust to the polar oceans. *Biogeosciences*, *13*(13), 3887–3900. https://doi.org/10.5194/bg-13-3887-2016

Saba, V. S., Friedrichs, M. A. M., Antoine, D., Armstrong, R. A., Asanuma, I., Behrenfeld, M. J., … Westberry, T. K. (2011). An evaluation of ocean color model estimates of marine primary productivity in coastal and pelagic regions across the globe. *Biogeosciences*, *8*(2), 489–503. https://doi.org/10.5194/bg-8-489-2011

Tagliabue, A., Aumont, O., DeAth, R., Dunne, J. P., Dutkiewicz, S., Galbraith, E., … Yool, A. (2016). How well do global ocean biogeochemistry models simulate dissolved iron distributions? *Global Biogeochemical Cycles*, n/a-n/a. https://doi.org/10.1002/2015GB005289

---

## Author Comment (AC3) · 18 Jul 2019

This paper presents a study evaluating the impact of Fe supply from Antarctic ice shelves and icebergs on productivity/chlorophyll in the Southern Ocean. It presents a thorough examination of the uncertainties associated with the fertilisation capacity of this input and highlights remaining differences between the observations and model results even when these Fe sources are included. The authors highlight particular areas where existing models can be improved, or futher in-situ observations are required. With some improvements, I believe this paper is a valuable addition to the field.

I am reviewing this paper with shallow knowledge of the biogeochemistry and will be focusing on iceberg and ice shelf melt.

**We thank reviewer #3 for his detailed review and general support for our manuscript.**
**We present our response in bold and preceded by '>' in case of formatting errors.**

Larger corrections
It was not clear whether the Fe supply is injected at a particular layer, and no further dynamics apply, or whether once the Fe is added, those waters are able to mix (as is likely to happen associated with the buoyancy injection from meltwater)? This applies throughout the paper, but in particular on page 13 (line 30-35) where you discuss the possible cause of differences between your primary production are that found in Laufkötter et al (2018). Some further discussion of this, and the general background associated with the meltwater pump would be valuable. Recent papers have shown the effect of this in Antarctic waters (St-Laurent et al., 2017, 2019: Cape et al., 2019) and in your discussion you only refer to this process associated with Greenland glaciers (pg 15, line 21). Similar to the meltwater pump model for ice shelves, are similar processes considered for iceberg melt? For iceberg melt occurring at depth, mixing with surrounding waters may result in upwelling of nutrient-rich waters, rather than the iceberg Fe-source remaining trapped below the ML.

**In our model configuration, the cavities below the ice shelves are not opened. To mimic the overturning circulation driven by these unresolved ice shelves, we used the parametrisation of Mathiot et al. (2017) which prescribes a meltwater flux of ice shelf uniformly distributed over the depth and width of the unresolved cavity opening, from the mean ice front draft down to the seabed, or the grounding line depth if it is shallower. Mathiot et al. (2017) showed that this parametrisation of the ice shelf melting drives a buoyant overturning circulation along the coast, i.e. the meltwater pump, similar to that simulated by cavities when they are explicitly resolved.**

**For icebergs, it is true that a similar mechanism may occur (Helly et al., 2011; Stephenson et al., 2011) but the scale of that process is small (Biddle et al., 2015). This subgrid-scale mechanism is not**

**represented in the iceberg and ocean models used to produce the iceberg meltwater climatology (Marsh et al., 2015; Merino et al., 2016) and not relevant with our model setup of 1° resolution. However, investigating different distribution of iceberg Fe fluxes allowed us to explore the potential impact of that mechanism on ocean biogeochemistry. The surface distribution of iceberg Fe fluxes can be seen as a highly effective meltwater pump, the homogeneous distribution throughout the water column as a moderate meltwater pump and a distribution at depth as an inefficient meltwater pump.**

**Regarding the study of Laufkötter et al. (2018), their method, results, and model outputs do not allow to disentangle physical or biogeochemical reasons for differences with our model. Many reasons might explain the very different sensitivity of C export to AIS Fe fluxes: different distribution of freshwater fluxes, different modelled physical properties, different nutrient distribution, a different relationship between primary productivity and C export (see also our answer to reviewer #1). In fact, a detailed and thorough comparison with that study is really challenging because we lack many information that would be necessary. These differences are really intriguing and would probably deserve a careful analysis involving a collaboration between the two groups.**

Smaller corrections

Abstract: Line 12-13: The comment that seasonal variations have regional impacts that are then "almost negligible" is slightly confusing. May be better to re-word this sentence?

**Sentence reworded as follows: "The Fe supply is effective all year round and seasonal variations of the iceberg Fe fluxes have regional impacts which are small for annual-mean primary productivity and C export at the scale of the SO"**

Pg2: Some other references to consider in this section are Cape et al (2019) (ice shelf meltwater pump), Biddle et al (2015), in-situ observations of productivity from iceberg melt

**References added in this section of the article.**

Line 17: I'm not sure you've defined AIS yet. Be very clear about the differences between AIS (I assume Antarctic Ice Sheet?), ice shelves and icebergs.

**The acronym AIS used for the Antarctic Ice Sheet is defined in the abstract.**

Line 27: "fueling" in what way? Is the Fe used, or is it just supplied?

**Here we mean "supplied".**
**Sentence modified.**

Line 34: remove "the" before "Prydz Bay"

**> Acknowledged and addressed**

Pg 3, line 18: I would read "along the water column" as along the iceberg tracks (spatial/ horizontal). Is this what you mean, or do you mean the vertical distribution?

**We mean vertical distribution, i.e. through the water column.**
**> Acknowledged and addressed**

Pg 4, line 10: For those unfamiliar with the model, a brief description here of how the freshwater fluxes are added would be helpful. Are the ice cavities simulated? Or is it a vertical wall in the model that freshwater/Fe is added through? In the latter case, what does "between the base and the grounding line of the ice shelves" then refer to – freshwater fluxes are equally added between the depth of the ice shelf (say 400 m) and the seabed? In this situation, many recent papers have shown that the strongest outflow is at the base of the ice shelf and diminishes with depth, in addition to buoyant upwelling to the surface (Naveira Garabato et al., 2017; Nakayama et al., 2014). Again, this is relevant to the meltwater pump.

**Please, see our answer to general concerns.**
**Text modified to detail that the parametrisation of the ice shelf melting from Mathiot el at. (2017) simulates the buoyant overturning circulation along the coast and the associated meltwater pump.**

Pg 6, line 24: "as well as in the Ross Sea until the Amundsen Sea" – I'm not sure what you mean by this? The Indian and Pacific sectors include these coasts? (See comment in figures about specifying what region you are referring to).

**You are totally right, Indian and Pacific sectors include these coasts.**
**We removed this part from the sentence.**
**The Southern Ocean sectors are added in Figure 1.**

Pg7, line 9-10: I am not sure what you mean by "Furthermore, in winter: : :".

**Sentence modified as follows:**
**"Furthermore, in winter, deep mixing entrained to the surface Fe that was released in summer below the euphotic zone and that escaped consumption by phytoplankton due to the lack of light."**

Pg 10, Lines 14-18: I think the meltwater pump should be included here – the ice shelf Fe is not just injected

deeper than the mixed layer.

**You are right. We modified the text in accordance.**

Line 33: "The mains" ! "The main"

**> Acknowledged and addressed**

Pg 11, Line 11: remove "the" in front of Bouvet.

**> Acknowledged and addressed**

Line 15: remove "by" in front of "1.3: : :"

**Replaced by « up to »**

Pg 13, Line 30-35: This deserves more discussion about why there are differences between the models with similar Fe fluxes. Are there physical differences in the models in how they treat mixing of meltwater/depth of meltwater input?

**Please see our answer to the general concerns.**

Pg 15, Line 20-23: This seems quite likely (e.g. Cape et al, 2019) – see earlier general comment. Line 34: "we did not explore"

Figures – I would like the labels on the maps for longitudes to be slightly larger, and to be consistent with the direction/order of labelling panels. You also refer to the different sectors a lot (e.g. Indian-Pacific sector) – is it possible to mark the boundaries of these sectors, perhaps just on the first figure?

**OK figures modified.**
**Indian, Atlantic, and Pacific sectors added in Figure 1.**

Figure 5 – what is the colorbar for this figure?

**The colour bar is identical to figure 4 and added in figure 5.**

Figure 8 has an incorrect caption (it is identical to Figure 7).

**Corrected to the right caption:**

**"Surface chlorophyll concentrations in summer (December, January, and February) from (a) satellite observations (MODIS-Aqua, Johnson et al. (2013)), (b) the CTL experiment, and (c) the SOLUB5 experiment in the Southern Ocean, south of 50° S.**

**References**

Biddle, L. C., Kaiser, J., Heywood, K. J., Thompson, A. F., & Jenkins, A. (2015). Ocean glider observations of iceberg-enhanced biological production in the northwestern Weddell Sea. *Geophysical Research Letters*, *42*(2), 459–465. https://doi.org/10.1002/2014GL062850

Helly, J. J., Kaufmann, R. S., Stephenson, G. R., & Vernet, M. (2011). Cooling, dilution and mixing of ocean water by free-drifting icebergs in the Weddell Sea. *Deep Sea Research Part II: Topical Studies in Oceanography*, *58*(11–12), 1346–1363. https://doi.org/10.1016/j.dsr2.2010.11.010

Johnson, R., Strutton, P. G., Wright, S. W., McMinn, A., & Meiners, K. M. (2013). Three improved satellite chlorophyll algorithms for the Southern Ocean. *Journal of Geophysical Research: Oceans*, *118*(7), 3694–3703. https://doi.org/10.1002/jgrc.20270

Laufkötter, C., Stern, A. A., John, J. G., Stock, C. A., & Dunne, J. P. (2018). Glacial Iron Sources Stimulate the Southern Ocean Carbon Cycle. *Geophysical Research Letters*, *45*(24), 13,377-13,385. https://doi.org/10.1029/2018GL079797

Marsh, R., Ivchenko, V. O., Skliris, N., Alderson, S., Bigg, G. R., Madec, G., … Zalesny, V. B. (2015). NEMO–ICB (v1.0): Interactive icebergs in the NEMO ocean model globally configured at eddy-permitting resolution. *Geoscientific Model Development*, *8*(5), 1547–1562. https://doi.org/10.5194/gmd-8-1547-2015

Mathiot, P., Jenkins, A., Harris, C., & Madec, G. (2017). Explicit representation and parametrised impacts of under ice shelf seas in the z$\pi$ coordinate ocean model NEMO 3.6. *Geoscientific Model Development*, *10*(7), 2849–2874. https://doi.org/10.5194/gmd-10-2849-2017

Merino, N., Le Sommer, J., Durand, G., Jourdain, N. C., Madec, G., Mathiot, P., & Tournadre, J. (2016). Antarctic icebergs melt over the Southern Ocean: Climatology and impact on sea ice. *Ocean Modelling*, *104*, 99–110. https://doi.org/10.1016/j.ocemod.2016.05.001

Stephenson, G. R., Sprintall, J., Gille, S. T., Vernet, M., Helly, J. J., & Kaufmann, R. S. (2011). Subsurface melting of a free-floating Antarctic iceberg. *Deep Sea Research Part II: Topical Studies in Oceanography*, *58*(11–12), 1336–1345. https://doi.org/10.1016/j.dsr2.2010.11.009